

# Technical note: Exploring parameter and meteorological uncertainty via emulation in volcanic ash atmospheric dispersion modelling

James M. Salter[1], Helen N. Webster[1,2], and Cameron Saint[2]

[1]Department of Mathematics and Statistics, University of Exeter, Exeter, EX4 4QF, UK
[2]Met Office, FitzRoy Road, Exeter, EX1 3PB, UK

**Correspondence:** James M. Salter (j.m.salter@exeter.ac.uk)

**Abstract.** Consideration of uncertainty in volcanic ash cloud forecasts is increasingly of interest, with an industry goal to provide probabilistic forecasts alongside deterministic forecasts. Simulations of volcanic clouds via dispersion modelling are subject to a number of uncertainties, relating to the eruption itself (mass of ash emitted, and when), parametrisations of physical processes, and the meteorological conditions. To fully explore these uncertainties through atmospheric dispersion model

simulations alone may be expensive, and instead an emulator can be used to increase understanding of uncertainties in the model inputs and outputs, going beyond combinations of source, physical and meteorological inputs that were simulated by the dispersion model. We emulate the NAME dispersion model for simulations of the Raikoke 2019 eruption, and use these emulators to compare simulated ash clouds to observations derived from satellites, constraining NAME source and internal parameters via history matching. We demonstrate that the effect of varying both meteorological scenarios and model parame-

ters can be captured in this way, with accurate emulation using only a small number of runs per meteorological scenario. We show that accounting for meteorological uncertainty simultaneously with other uncertainties may lead to the identification of different sensitive model parameters, and may lead to less constrained source and internal NAME parameters, however through idealised experiments we argue that this is a reasonable result and is properly accounting for all sources of uncertainty in the model inputs.

## 1 Introduction

Atmospheric dispersion models are used to predict the atmospheric transport, dispersion and removal of ash emitted during a volcanic eruption. NAME (Numerical Atmospheric-dispersion Modelling Environment, Jones et al. (2007); Beckett et al. (2020)) is the atmospheric dispersion model used by the London VAAC (Volcanic Ash Advisory Centre), and is used during Icelandic eruptions (such as the 2010 eruption of Eyjafjallajökull, Webster et al. (2012)) to provide guidance on the presence

of ash in the atmosphere, and reduce the risk to aviation. Uncertainties exist in dispersion model predictions due to errors and uncertainties in the input meteorological data, the model parametrisations (describing physical processes such as turbulence) and estimates of volcanic ash emissions. It is important to understand and to quantify these uncertainties, and to communicate





uncertainty information to end users. Indeed, there is an aim for the VAACs to be able to provide probabilistic forecasts, alongside deterministic volcanic ash cloud forecasts.

When an expensive model cannot be evaluated fast or often enough at varied settings of its inputs, it can be replaced by an 'emulator', trained on a relatively small number of simulations of the true physical model. This emulator can then be used to perform analyses that would not be possible (within a reasonable timeframe) using the original model. An emulator allows fast predictions to be evaluated for unseen combinations of the model inputs, allowing the (often) high-dimensional input space to be more extensively explored, with all sources of uncertainty accounted for. An emulator of the true model can then be used for

tasks including sensitivity analysis (Saltelli et al., 1999; McNeall et al., 2023), Bayesian calibration (Kennedy and O'Hagan, 2001; Higdon et al., 2008; Sexton et al., 2011) and history matching (Craig et al., 1996; Vernon et al., 2010; Williamson et al., 2013; Andrianakis et al., 2015; Salter et al., 2019).

Within dispersion modelling, past studies have taken different approaches to assessing uncertainties in the ash cloud forecast produced by NAME for volcanic eruptions, exploring the effect and importance of different model inputs, both with and

without emulation (e.g., Leadbetter et al. (2022); Jones et al. (2023)). Harvey et al. (2018) studied the 2010 Eyjafjallajökull eruption with NAME, and emulated the mean ash column loading for 75 geographical regions, for several hourly timepoints. NAME was run at two spatial resolutions, with the faster, coarser version being used to inform emulators for the slower version. This study considered sensitivity of the output to the source parameters (those describing the emissions) and internal NAME input parameters via emulators, but did not calibrate to satellite observations, and all simulations used a single deterministic

meteorological scenario, i.e. uncertainty in the meteorological conditions was not considered.

For the Raikoke 2019 eruption, Capponi et al. (2022) ran simulations of NAME with varied source and internal input parameters, and with input meteorology drawn from an 18 member ensemble, comparing the NAME ash cloud forecasts to satellite observations, constraining the source and internal inputs as the eruption progressed in time in order to find the model parameters that lead to accurate (or most accurate) output. Unlike Harvey et al. (2018), emulation was not used: instead multiple

batches of NAME simulations were performed, and only these simulations were used to produce estimates of the inputs given the satellite observations.

In this article, we emulate NAME for simulations of the 2019 Raikoke eruption, and combine aspects of the two approaches described above: fitting emulators to output summaries as in Harvey et al. (2018), and comparing to observations as in Capponi et al. (2022). Unlike Harvey et al. (2018), we vary the meteorology and account for this source of uncertainty via emulators.

Unlike Capponi et al. (2022), we use different metrics for comparing NAME and observations, and calibrate the inputs using emulators, rather than constraining input distributions using only the available set of NAME simulations. This allows the full joint space of NAME internal parameters, source inputs, and meteorological scenarios to be explored and constrained, with all uncertainties accounted for, aiming to protect against incorrectly over-constraining inputs due to not considering all sources of uncertainty in the inputs, or due to only considering a limited number of NAME simulations.

Given emulators for the ash cloud at different lead times and different spatial regions, we constrain the space of internal and input model parameters using satellite retrievals of ash column load, giving posterior estimates of model parameters via history matching. We consider different ways of emulating NAME, and consider different metrics for calibrating the parameters whilst





accounting for meteorological uncertainty, using NAME simulations at known inputs as proxy observations as a proof-of-concept, before using satellite retrievals from the Raikoke eruption. At longer lead times, or as we restrict to a smaller subset of the output, the meteorological scenario has a larger impact, and accounting for this properly can highlight different parameter sensitivities and relationships, through a more rigorous exploration of the uncertainties due to all variable inputs.

Section 2 describes the NAME model, the simulation inputs and outputs used in this study, and the available satellite observations for the Raikoke 2019 eruption. Section 3 outlines emulation and history matching, and how these are applied to NAME. Section 4 fits emulators to different summaries of the NAME output, assesses their validity for predicting out-of-sample, and uses these emulators to calibrate the uncertain input parameters, where the observations are either a known simulation of NAME or derived from satellite retrievals. Section 5 discusses the implications of the results, and suggests potential extensions.

## 2 Volcanic ash simulations and observations

### 2.1 Modelling volcanic ash with NAME

NAME is an offline atmospheric dispersion model driven by input meteorology (Jones et al., 2007). In the Lagrangian framework, model particles, each representing a certain mass of volcanic ash, are advected through the model atmosphere, according to the ambient wind obtained from the input meteorological data. Dispersion of volcanic ash is simulated using random walk techniques. Removal of ash from the atmosphere by wet and dry deposition processes are parameterised within NAME, including gravitational settling of heavy ash particles. Details of the emissions, such as the emission rate, the emission height, the emission time and the particle size distribution, need to be specified by the user.

Typically Numerical Weather Prediction (NWP) data is used as input meteorological data for NAME. Meteorological uncertainty is generally considered by means of meteorological ensembles - a set of NWP forecasts, referred to as ensemble members, obtained by running NWP models multiple times with perturbed initial conditions. The spread in the ensemble forecasts represents the meteorological uncertainty.

### 2.2 Dispersion modelling of Raikoke 2019

Raikoke is a volcanic island located at 48.29°N, 153.25°E. The 2019 eruption studied here began around 1800 UTC on 21st June 2019, and lasted for approximately 12 hours. Aspects of this event have been extensively studied, including by de Leeuw et al. (2021); Smirnov et al. (2021); Capponi et al. (2022); Harvey et al. (2022); Prata et al. (2022). For this study, we use an ensemble of 1000 NAME simulations of the Raikoke eruption performed by Capponi et al. (2022).

The eruption source parameters (height, distal fine ash fraction, mass eruption rate, ash density, and duration) and internal NAME inputs (relating to free tropospheric turbulence and unresolved mesoscale motions (Webster et al., 2018)) that were varied in these 1000 simulations, and their chosen prior ranges, are shown in Table 1. The 1000 sets of inputs were chosen via a Latin hypercube design, so that the design is space-filling across the full parameter space. For full details of the chosen prior ranges and other design choices made when generating the initial 1000 member design, see Capponi et al. (2022).





**Table 1.** List of source parameters and internal NAME parameters that are varied, and their prior ranges, in the NAME simulations of the Raikoke 2019 eruption.

| Parameter | Description | Range |
|---|---|---|
| $H$ | Plume height above summit | $[9, 17]$ km |
| $DFAF$ | Distal fine ash fraction | $[0.5, 20]$ % |
| $MER_F$ | Uncertain factor for Mass Eruption Rate (MER) | $[\frac{1}{3}, 3]$ |
| $\rho$ | Ash density | $[1350, 2500]$ kg m$^{-3}$ |
| $L$ | Eruption duration | $[9, 15]$ hours |
| $\sigma_u$ | Standard deviation of horizontal velocity for free tropospheric turbulence | $[0.0025, 2.71]$ m s$^{-1}$ |
| $\sigma_w$ | Standard deviation of vertical velocity for free tropospheric turbulence | $[0.001, 1]$ m s$^{-1}$ |
| $\tau_u$ | Horizontal Lagrangian timescale for free tropospheric turbulence | $[100, 900]$ s |
| $\tau_w$ | Vertical Lagrangian timescale for free tropospheric turbulence | $[33\frac{1}{3}, 300]$ s |
| $m\sigma U$ | Standard deviation of horizontal velocity for unresolved mesoscale motions | $[0.27, 1.74]$ m s$^{-1}$ |

The Mass Eruption Rate (MER) is an important driver of the model output and is calculated from the other inputs as:

$$MER = 50.7 \times 10^7 \times H^{1/0.241} \times MER_F \times DFAF,$$

where the $MER_F$ and $DFAF$ inputs have been applied to the relationship from Mastin et al. (2009), written here in g h$^{-1}$.

For each of the 1000 sets of source and internal parameters, the meteorological input is sampled from an 18 member meteorological ensemble MOGREPS-G (members labelled 0-17), initialised at 1200 UTC on 21st June 2019 (Bowler et al., 2008). Each input set is simulated with a single meteorological scenario. The resulting 1000 member NAME ensemble contains 30 simulations with meteorological scenarios 0 and 17, and 58-60 simulations with scenarios 1-16. Capponi et al. (2022) performed subsequent simulations based on their constrained parameter distributions, however we restrict to the initial 1000-member space-filling design.

## 2.3 Observations

To enable simulations of the event to be compared to the real-world, observations from the geostationary satellite Himawari-8 are used (Bessho et al., 2016). This data is available at both high temporal and high spatial resolution. Pixels are classified as either containing ash or not with the detection algorithm described in Saint (2023). The volcanic ash retrieval algorithm of Francis et al. (2012) is used to determine an estimate of ash column loading for the pixels classified as containing ash, together with an error estimate on the retrieved values. Further processing is undertaken to additionally classify pixels as clear sky and





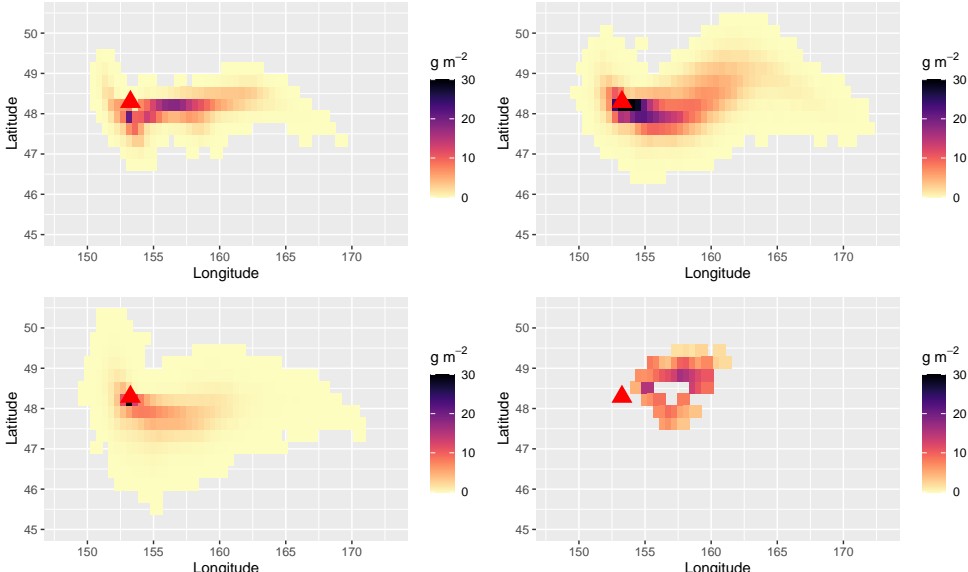

**Figure 1.** Predicted ash column loads (in $g\,m^{-2}$) at 0600 UTC on 22/06/2019 from 3 NAME runs with varied source, internal and meteorological inputs, and the corresponding satellite retrieval estimates (bottom right panel). The red triangle indicates the location of Raikoke.

regrid to a regular latitude-longitude grid corresponding to that used for the NAME predictions, with resolution $0.45° \times 0.3°$.
This processing is as described in Pelley et al. (2021) except the target grid is slightly higher resolution and there is no averaging in time, with only the satellite data from the times indicated used.

  The observed ash cloud may be incomplete, due to failures of the detection or retrieval algorithm to either classify pixels as ash or reach an acceptable solution for the ash cloud properties. In this case, the observed plume will consist of a subset of the modelled plume, as seen in Figure 1, which compares the output of 3 NAME simulations to satellite retrievals, 12 hours after
110 the start of the eruption. Note that the observations plotted here are not perfectly consistent with Capponi et al. (2022), due to use of an updated method for processing satellite observations and regridding of the observations.

## 3 Methods

For clarity around the use of 'model', hereafter 'simulator' will refer to NAME, and 'emulator' will refer to a statistical model.

### 3.1 Emulation

Let the physics-based simulator (i.e., NAME) be denoted by $f(\cdot)$, which takes a vector of inputs $\mathbf{x} \in \mathcal{X} \subset \mathbb{R}^p$, where $\mathcal{X}$ denotes the $p$-dimensional space of possible simulator inputs, and returns output $f(\mathbf{x})$. Here, $\mathbf{x}$ represents the values of the inputs described in Table 1 (i.e., the source inputs and internal NAME parameters), with the meteorological input ignored for now. The simulator is run for a set of $n$ inputs $\mathbf{X} = (\mathbf{x}_1, \ldots, \mathbf{x}_n)$ for $\mathbf{x}_i \in \mathcal{X}$, resulting in a set of $n$ simulations $\mathbf{F} = (f(\mathbf{x}_1), \ldots, f(\mathbf{x}_n))$,





where each $f(\cdot) \in \mathbb{R}^\ell$ is vectorised output of (or processed from) a single NAME simulation. Given a set of simulations, we fit a statistical model or 'emulator', so that the simulator output can be predicted for any new input, without having to run the true, expensive, simulator.

When the simulator output is a single value (or has been summarised to be a single value, for example, total mass of atmospheric ash at 0600UTC on 22/06/2019), i.e., $\ell = 1$, Gaussian Processes (GPs) are a natural choice of emulator, allowing fast prediction at unseen settings of $\mathbf{x}$, along with uncertainty on this prediction (we have not observed the true simulator at these points), which generally increases as we move further (in input space) from known data points. In general, conditional on the $n$ simulator runs $\mathbf{F}$, we emulate $f(\mathbf{x})$ as:

$$f(\mathbf{x})|\mathbf{F} \sim GP(\mu(\mathbf{x}), C(\mathbf{x}, \mathbf{x}')). \tag{1}$$

for mean function $\mu(\mathbf{x})$ (which may range from constant to a complex regression model in the inputs) and covariance function $C(\mathbf{x}, \mathbf{x}')$ (often squared exponential or Matérn), containing parameters controlling the variance, and correlation in each input dimension. The covariance may also include a 'nugget' term (Andrianakis and Challenor, 2012), representing variance around training points. If this is zero, then the GP interpolates training data exactly. If this is estimated, it may represent a number of uncertainties, including the effect of unaccounted for (inactive) inputs or stochasticity in the simulator. The nugget may be assumed to be constant across input space, or dependent on $\mathbf{x}$ (e.g., hetGP (Binois et al., 2018)).

When $\ell > 1$, e.g., if we wish to emulate spatial, temporal, or spatio-temporal simulator output, the two main approaches are based around GPs: assuming independence and fitting GP emulators to each output independently (Lee et al., 2012), or using a basis decomposition over the output $\mathbf{F}$ and emulating the coefficients given by projection onto this basis with GPs (Higdon et al., 2008; Salter et al., 2019). In what follows, we consider total ash column loads, aggregated across different regions and for different lead times (as in Harvey et al. (2018)), and fit independent GP emulators to each of these summaries of the simulator output.

NAME also requires meteorological data as an input, with meteorological scenario $m$ drawn from $\mathcal{M}$, some space of possible meteorologies; in this application, $\mathcal{M} = \{0, \ldots, 17\}$ representing the 18 MOGREPS-G ensemble members for which simulations were run by Capponi et al. (2022). Given the nature of the NAME simulations in this study, we can account for this meteorological uncertainty via emulation in two different ways.

First, we assume that NAME is stochastic with respect to $m$ (i.e. if a NAME simulation is run at input $\mathbf{x}$ for different choices of $m$ from $\mathcal{M}$, the output $f(\mathbf{x})$ is different). We fit a single emulator, that captures the dependence due to the inputs $\mathbf{x}$, and the variability across the different meteorological scenarios $m$. This can be approached via equation (1), with the nugget term representing the variability in the output at $\mathbf{x}$ due to $m$. Rather than exactly emulating any combination of $(\mathbf{x}, m)$, this emulates the mean of NAME at $\mathbf{x}$, given any sample of $m$ from the 18-member MOGREPS-G ensemble.

For a particular $\mathbf{x}$, we can write $\mathrm{E}[f(\mathbf{x})], \mathrm{Var}[f(\mathbf{x})]$ (we don't need to know $m$, because we are not emulating or predicting for a specific $m$, but across all $m$). The expectation is therefore targeting the mean of the NAME output at $\mathbf{x}$, if we were to run it at all 18 meteorological inputs. The variance here accounts both for extrapolation from observed points in the input space, but also for the fact that we can view $f(\mathbf{x})$ at 18 different choices of $m$.




As a second approach, because the meteorological scenarios are sampled from a small, finite set, each of the 18 meteorological scenarios $m$ could be thought of as representing a different configuration, $f_m$, of NAME. Each configuration gives

deterministic output at input $\mathbf{x}$, and we could emulate each in turn:

$$f_m(\mathbf{x})|\mathbf{F}_m \sim GP(\mu_m(\mathbf{x}), C_m(\mathbf{x}, \mathbf{x}')), \quad m = 0, \ldots, 17, \tag{2}$$

where $\mathbf{F}_m$ is the subset of $\mathbf{F}$ containing simulations run using meteorological input $m$, and the mean $\mu_m(\cdot)$ and covariance $C_m(\cdot, \cdot)$ functions are estimated for each $m$ separately.

In this approach, for a particular summary of the NAME output, we fit a set of 18 emulators, and for any choice of $\mathbf{x}$, we can

predict the simulator output at this input vector for all of the 18 meteorological scenarios:

$$(\mathrm{E}[f_m(\mathbf{x})], \mathrm{Var}[f_m(\mathbf{x})]), \quad m = 0, \ldots, 17.$$

The variance here only reflects the fact that we have not simulated NAME itself at all combinations of $(\mathbf{x}, m)$ (the emulator is extrapolating from known points).

In summary, the first approach predicts the output of NAME at $\mathbf{x}$, averaged across the 18 meteorological ensemble members,

and the second predicts the output for a specific combination of $(\mathbf{x}, m)$. The latter has the benefit that it represents NAME in a truer sense: we can query the emulator for any combination of $(\mathbf{x}, m)$, as with NAME itself. However, the first approach does not in general require a small ensemble of meteorological scenarios, and could instead capture the uncertainty due to a more continuous set of scenarios (e.g., if every simulation of NAME was performed with a slightly perturbed meteorological input).

In Section 4 we consider both approaches, referring to them as the 'Overall' and 'MET-specific' emulators respectively.

**3.2 History matching**

Given an emulator(s), we may wish to compare to real-world observations of the modelled quantity, and constrain the input space based on this. Often this is done via history matching (Craig et al., 1996), where we rule out clearly poor choices of the inputs, returning a space of not implausible $\mathbf{x} \in \mathcal{X}$, with the retained part of $\mathcal{X}$ containing inputs that may be consistent with observations, given uncertainty due to emulation, observation error, and inconsistencies between the real-world and the

175 simulator. We assume:

$$z = f(\mathbf{x}^*) + e + \eta,$$

where $f(\mathbf{x}^*)$ is the simulator at 'best' input $\mathbf{x}^*$, which represents real-world observations $z$ up to observation error $e$ and model discrepancy $\eta$ (differences between observations and the simulator independent of the inputs $\mathbf{x}$, e.g., due to missing or simplifications of physical processes). The simulator $f(\cdot)$ can be replaced by an emulator, and the 'implausibility' of an input

$\mathbf{x}$ is given by (Vernon et al., 2010; Williamson et al., 2013):

$$\mathcal{I}(\mathbf{x}) = \frac{|z - \mathrm{E}[f(\mathbf{x})]|}{\sqrt{\mathrm{Var}[f(\mathbf{x})] + \mathrm{Var}[e] + \mathrm{Var}[\eta]}}, \tag{3}$$





where the expectation and variance of $f(\mathbf{x})$ are from an emulator (if we were able to run the simulator at any $\mathbf{x}$, $\mathrm{Var}[f(\mathbf{x})] = 0$), and $\mathrm{Var}[e], \mathrm{Var}[\eta]$ are the variance of the observational error and model discrepancy respectively. The 'Not Ruled Out Yet' (NROY) space is defined as the subset of $\mathcal{X}$ where the implausibility falls below threshold $T$:

$$\mathcal{X}_{NROY} = \{\mathbf{x} \in \mathcal{X} | \mathcal{I}(\mathbf{x}) < T\},$$

with $T = 3$ a standard choice by the Three Sigma Rule (Pukelsheim, 1994).

$\mathcal{X}_{NROY}$ contains settings of the inputs where, given all sources of uncertainty, we cannot say that output $f(\mathbf{x})$ is inconsistent with the observed value, $z$. It is not guaranteed that such an $\mathbf{x}$ produces simulator output close to $z$, due to emulator variance, but that we are not yet sure if it will. Initially, it may not be possible to rule out some parts of parameter space due to high emulator variance (perhaps due to lack of training points), but we may be able to later rule out these regions by running new designs on the expensive simulator and refining emulators.

History matching is often used as an iterative procedure in this way, with the goal to zoom in on parts of $\mathcal{X}$ with output consistent with $z$, and can be performed for multiple outputs or summaries simultaneously (Vernon et al., 2010; Williamson and Vernon, 2013). The implausibility for all variables may be required to fall below $T$, but alternatively a more conservative definition may be used, with $\mathbf{x}$ instead ruled out based on the $k^{th}$ highest implausibility at $\mathbf{x}$, to protect against incorrectly discarding an input due to a single poor emulator, or due to misspecified discrepancy. If emulating the entire plume rather than summaries of the output, so that the observations and emulator prediction $\mathrm{E}[f(\mathbf{x})]$ are $\ell$-length vectors, the formula in (3) generalises to a distance metric across $\ell$-dimensional space, with $\ell \times \ell$ observation error and discrepancy variance matrices. Full details, including how to efficiently history match for high $\ell$, are provided in Salter and Williamson (2022).

## 3.3 History matching for NAME

In Section 4, we compare history matching for NAME modelling of the Raikoke eruption using the NAME simulator only, using different choices of emulator, and different levels of conservatism. The overall goal is not to calibrate $m$: these are deemed to be possible representations of the true meteorological conditions. The aim is to calibrate the inputs relating to NAME and the eruption itself (i.e., $\mathbf{x}$), whilst accounting for the meteorological uncertainty.

In this section, $\mathcal{I}(\mathbf{x})$ (no subscript) denotes implausibility calculated from the single emulator fitted across all meteorological scenarios (e.g. (1)), and $\mathcal{I}_m(\mathbf{x})$ denotes the implausibility at $\mathbf{x}$ for meteorology $m \in \mathcal{M}$, calculated via emulators (2). For any given $\mathbf{x}$, we can efficiently evaluate the expectation and variance for $f_m(\mathbf{x})$ for each $m$ using the emulators, and hence calculate an implausibility for each choice of $m$, $\{\mathcal{I}_0(\mathbf{x}), \dots, \mathcal{I}_{17}(\mathbf{x})\}$. By repeatedly sampling $\mathbf{x} \in \mathcal{X}$, we can use this to explore all possible combinations of $(\mathbf{x}, m)$ and account for uncertainties in both inputs $\mathbf{x}$ and meteorological scenario $m$ simultaneously, assessing how consistent each combination of $(\mathbf{x}, m)$ is with observations.

We outline the rationale behind each choice of metric here. The first two definitions rely on a random choice of $m$, and should be broadly similar (up to using NAME vs using an emulator for NAME), whereas the latter two account for the meteorological uncertainty via different emulators. Both aim to ensure we do not rule choices of $\mathbf{x}$ that may in fact be correct if we were to simulate $f(\mathbf{x})$ for an alternative meteorological scenario from the MOGREPS-G ensemble.





**Option 1: Simulator-only $\mathcal{X}_S$**

As a benchmark to compare the emulation approaches to, we define an NROY space, $\mathcal{X}_S$, using only the 1000 NAME sim-
ulations (a similar approach to Capponi et al. (2022), but with different summaries). Therefore, in (3), we can only calculate
$\mathcal{I}(\mathbf{x})$ at the 1000 input vectors $\mathbf{x}$ NAME has been simulated at, with $\mathrm{E}[f(\mathbf{x})] = f(\mathbf{x})$ and $\mathrm{Var}[f(\mathbf{x})] = 0$. The NAME output
is dependent on the particular meteorological scenario each $\mathbf{x}$ was run with, so that the outputs that are discarded for being

too dissimilar from the observations, and hence the posterior distributions over $\mathbf{x}$, may be biased by which $m$ was randomly
sampled for each simulation of NAME.

   If uncertainty in the NAME output due to meteorology is small (relative to observation error) this may not matter, however
there will always be a risk that a particular simulation $f_m(\mathbf{x})$ is unrepresentative of simulations at $\mathbf{x}$ for some different $m$, and
hence this definition can't completely capture meteorological uncertainty in the output.

**Option 2: Pseudo-simulator $\mathcal{X}_P$**

Option 1 is restrictive due to requiring simulations of NAME, however we can approximate what this NROY space would be
for other choices of $\mathbf{x}$ via emulation. To represent this approach, we sample $\mathbf{x} \in \mathcal{X}$, sample a meteorological scenario $m \in \mathcal{M}$
uniformly, evaluate the emulator $f_m(\mathbf{x})$ for this combination of $(\mathbf{x}, m)$, and hence calculate $\mathcal{I}_m(\mathbf{x})$. This is equivalent to defining
NROY space without accounting for the meteorological uncertainty, and is akin to the approach in Capponi et al. (2022).

To remove the effect of randomly sampling $m$, this space can be found by evaluating $\mathcal{I}_m(\mathbf{x})$ across all $m \in \mathcal{M}$, and keeping
$\mathbf{x}$ such that it is not inconsistent with observations for at least half of the meteorological scenarios, i.e.:

$$\mathcal{X}_P = \{\mathbf{x} \in \mathcal{X} | \mathcal{I}_m(\mathbf{x}) < T \text{ for } \geq 9/18 \, m \in \mathcal{M}\}.$$

If $\mathbf{x}$ is considered not implausible for only $\leq 8$ of the meteorological scenarios, it is therefore more likely we would have run
NAME for this $\mathbf{x}$ at one of the 10 implausible choices, and therefore rule this $\mathbf{x}$ as it is inconsistent with reality.

**Option 3: Overall $\mathcal{X}_O$**

Here, we evaluate the overall emulator (i.e. treating NAME as stochastic with respect to $m$, so that the effect of $m$ is accounted
for within this single emulator) and hence we have a single implausibility per $\mathbf{x}$ that captures all uncertainties, including due to
choice of meteorological scenario:

$$\mathcal{X}_O = \{\mathbf{x} \in \mathcal{X} | \mathcal{I}(\mathbf{x}) < T\}.$$

**Option 4: Conservative $\mathcal{X}_C$**

If a parameter choice $\mathbf{x}$ is implausible for 1 or several meteorological scenarios, it may still lead to acceptable output if we ran
NAME for a different $m$, and so we may wish to retain that $\mathbf{x}$. Therefore, we should only rule out parts of $\mathcal{X}$ if all meteorological
ensemble members lead to implausible simulator output at $\mathbf{x}$. Conversely, we should keep any input $\mathbf{x}$ if it is deemed to be non-
implausible for at least 1 meteorological ensemble member (whether this is because this combination of $(\mathbf{x}, m)$ leads to NAME





being consistent with the observations, or because the emulator variance is high enough that we cannot safely say it doesn't match). Therefore, here we define NROY space such that we retain $\mathbf{x} \in \mathcal{X}$ if $\mathcal{I}_m(\mathbf{x})$ falls below the threshold for any $m \in \mathcal{M}$:

$$\mathcal{X}_C = \{\mathbf{x} \in \mathcal{X} | \mathcal{I}_m(\mathbf{x}) < T \text{ for any } m \in \mathcal{M}\},$$

where the implausibility $\mathcal{I}_m(\mathbf{x})$ is evaluated using the emulator for $f_m(\mathbf{x})$.

## 4 Results

In this section, we emulate aspects of NAME output to allow a more extensive exploration of the effects that the source and internal inputs, $\mathbf{x}$, and the meteorological input, $m$, have on the output, via the cheap-to-evaluate GP emulators. We fit emulators for different lead times and regional aggregations (Section 4.1), and assess whether these are accurate for NAME runs not used in the training set (Section 4.2), and how this changes as the time since the start of eruption increases, or for more localised subsets of the output. We show that the emulators can be used to consider the distribution of output at $\mathbf{x}$ across all possible

$m$ (Section 4.3) and use this to motivate the importance of accounting for meteorological uncertainty when calibrating $\mathbf{x}$. We assess the accuracy of different emulators and definitions of NROY space in an idealised setting to demonstrate performance (Section 4.4) before considering the satellite observations (Section 4.5).

### 4.1 Emulating summaries

Using the ensemble of 1000 NAME runs simulating the Raikoke eruption as described in Section 2.2, and the emulation and
calibration techniques described in Section 3, we emulate several summaries of the full NAME output. So that comparisons with observations are possible, we subset the output to grid locations at which satellite observations are available for that timepoint, and aggregate the ash column loads across these locations only (or subsets of these). We consider 3 particular timepoints (12, 24 and 36 hours after the eruption started, referred to as T1, T3 and T5 respectively for consistency with Capponi et al. (2022)), for different subsets of the observed plume:

1. Total airborne ash at T1, T3 and T5

   2. Total airborne ash at T1 split into North (N) vs South (S) by latitude = 48.1°N

   3. Total airborne ash at T1 split into West (W) vs East (E) by longitude = 157.7°E

   4. Total airborne ash at T1 split into 4 regions (NW, NE, SE, SW) given by latitude = 48.1°N and longitude = 157.7°E

Considering totals at different forecast times will demonstrate how the meteorological variability changes over time, and
whether emulation accuracy changes with this. Splitting into different geographical regions at T1 demonstrates whether spatial variability can be captured, and whether there are different effects of $\mathbf{x}$ and $m$ as we consider smaller regions. This will also assess whether meteorological dependence matters at shorter forecast lead times.



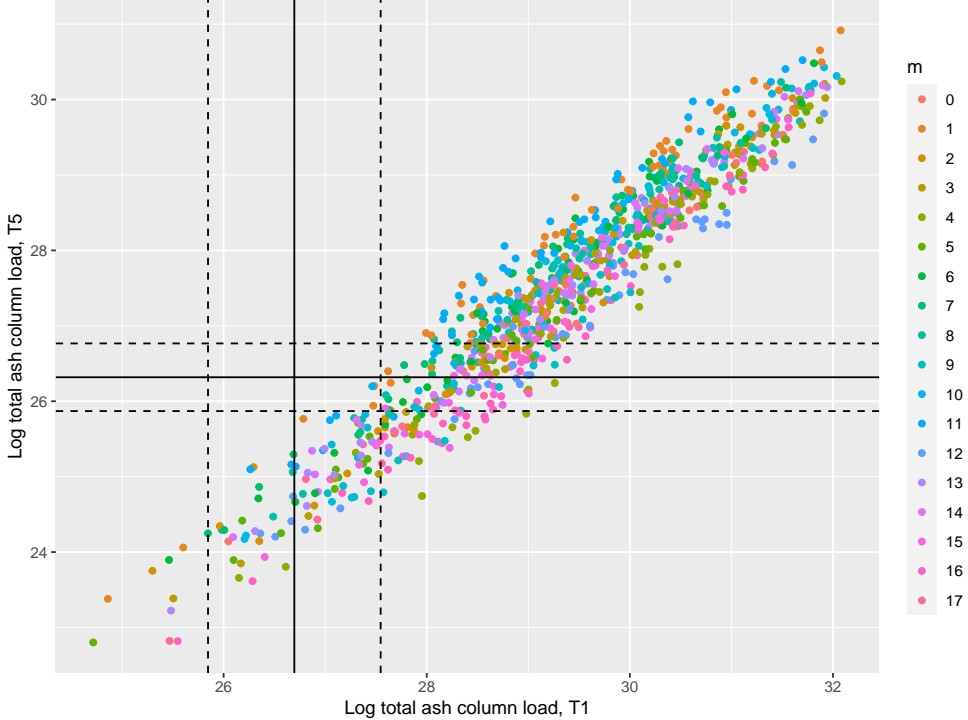

**Figure 2.** (Log) total ash column load (in $g$) for the 1000 NAME simulations at T1 and T5, aggregated across locations where ash was observed, with points coloured by the meteorological scenario used to run the simulation. The vertical and horizontal lines indicate the median (solid line) and 99% uncertainty interval (dashed lines) from the satellite observations for these 2 timepoints.

Satellite observations are given at grid box level, with median, 10th and 90th percentiles. We perform the same aggregations as with the NAME output to allow for comparisons between modelled ash and reality. To estimate uncertainty on these totals, we sum the 10th and 90th percentiles for the relevant grid boxes, and assuming Normality, calculate the variance, $\mathrm{Var}[e]$. This is an overestimate as it assumes perfect correlation and hence inflates this variance, however the true correlation structure is unknown, and this conservative estimate is preferable to underestimation (e.g., through assuming independent errors).

Figure 2 plots the log totals (across observed ash locations) across the 1000 runs for T1 and T5, coloured by which MOGREPS-G member was used as the meteorological scenario for each simulation, with the solid vertical and horizontal lines showing the median satellite observation for each metric, and the dotted lines showing 99% intervals. Figure 3 plots the split into north and south (left) and west and east (right) at T1, with more variability in the output when split into north and south, and a clear effect of $m$. Figure 4 similarly plots the output for the split into 4 regions, for the NE and SE regions (left) and the NW and SW regions (right). There is a positive but relatively noisy relationship between the regions in each case (and again a suggestion that the meteorology drives shifts in this relationship). In each case, there are NAME simulations that lie within the 99% distribution around the satellite observations for the pair of metrics plotted. The observed column loads are at the lower end of the 1000 NAME simulations, which is a function of the choice of prior ranges for the NAME inputs (Table 1).

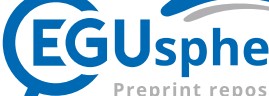

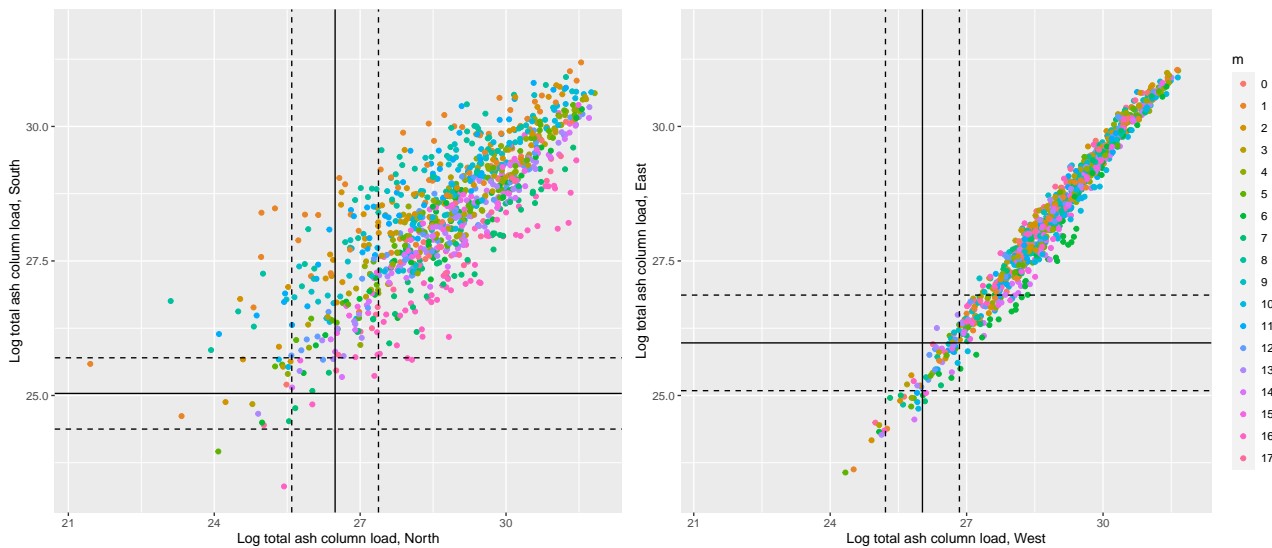

**Figure 3.** (Log) total ash column load (in $g$) for the 1000 NAME simulations, for the split at T1 into north and south regions (left), and the split at T1 into west and east regions (right), coloured by the meteorological scenario used to run the simulation. Median observations (solid lines) and 99% uncertainty (dotted lines) are plotted for each regional summary.

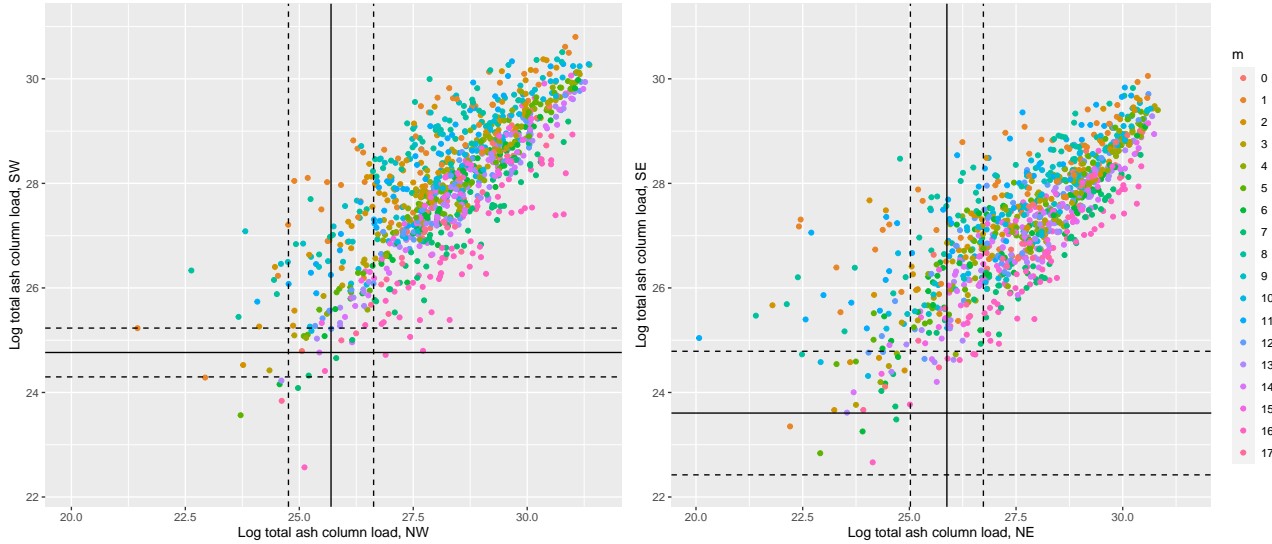

**Figure 4.** (Log) total ash column load (in $g$) for the 1000 NAME simulations, for the split at T1 into 4 regions (NW/SW plotted on the left, NE/SE plotted on the right), with points coloured by the meteorological scenario used to run the simulation. Median observations (solid lines) and 99% uncertainty (dotted lines) are plotted for each regional summary.





For each NAME summary above, we fit the emulators described in Section 3.1, i.e. 1) fitting a single emulator and capturing the meteorological uncertainty within this ('overall' emulator), and 2) separate emulators trained for each meteorological ensemble member in turn ('MET-specific' emulator). In each case, we emulate the logarithm of the total ash column load, and

also use the log of the mass eruption rate (MER) as part of the input vector $\mathbf{x}$, as ash column loads are linearly related to the emission rate.

For consistency, we split the 1000 NAME simulations into the same training (75%) and validation (25%) sets every time, stratified by $m$. The overall emulator is fitted to these 750 training points, whilst the MET-specific emulators are trained on 22 points (MOGREPS-G ensemble members 0 and 17), or 44-45 points (all other ensemble members). In all subsequent sections,

predictions for the points in the validation set are directly comparable as the same runs are always removed.

## 4.2   Out-of-sample predictions

To assess emulator accuracy for the different outputs, we evaluate each emulator for the 250 points in the validation set, and compare these predictions to the true NAME output. In all cases, around 95% of true values lie within 95% prediction intervals, indicating that the emulators are capturing uncertainty and predicting out-of-sample well. Figures 5 and 6 plot the NAME

output against emulator predictions (mean and 95% prediction intervals) for the 250 validation points, for the totals at T1 and T5 respectively, for the MET-specific (left) and overall (right) emulators (see Appendix for the other emulated summaries).

The error bars in the MET-specific plots account only for extrapolation via the emulator to combinations of $(\mathbf{x}, m)$ that were not included in the training set. The error bars in the 'Overall' plots account for both not training at a particular $\mathbf{x}$, and also the uncertainty across all meteorological scenarios. These predictions always exhibit larger uncertainty, despite the fact that the

'Overall' emulator is being trained with 750 points, whereas the individual $m$-dependent emulators are trained with at most 45 points. This indicates that given $m$, dependence due to the other inputs $\mathbf{x}$ is predictable.

Similarly, in each case the mean predictions lie closer to the diagonal in the MET-specific cases, because here we are plotting an emulator prediction for a specific $m$ against the NAME output at that $(\mathbf{x}, m)$. For the 'Overall' plots, this compares a prediction for $f(\mathbf{x})$, i.e. averaged across all meteorological scenarios, against the output for a specific $m$. If there is variability

between running NAME for different meteorological scenarios, then the mean prediction should appear less consistent with particular NAME runs. However, as long as the particular output is generally captured within the uncertainty around this emulator prediction (which is the case around 95% of the time), then this is not an issue and the emulator is capturing the uncertainty due to $m$ as intended.

For T1, both emulators exhibit strong agreement between the emulator mean and the true NAME output. The strong corre-

lation in the 'Overall' plot is therefore indicating that there is low variability in the T1 total due to the meteorological scenario (mean prediction across all $m$ is fairly consistent with the NAME output generated by a randomly-sampled $m$). At longer forecast lead times, we would expect that the meteorological ensemble diverges and causes greater uncertainty at a given $\mathbf{x}$, and this is demonstrated by predictions at T5 (Figure 6). The MET-specific emulators are still quite accurate (indicating that when we explicitly account for both $\mathbf{x}$ and $m$, the output is predictable), but there are larger error bars for the overall emulator,

indicating that there is now a larger effect due to $m$.




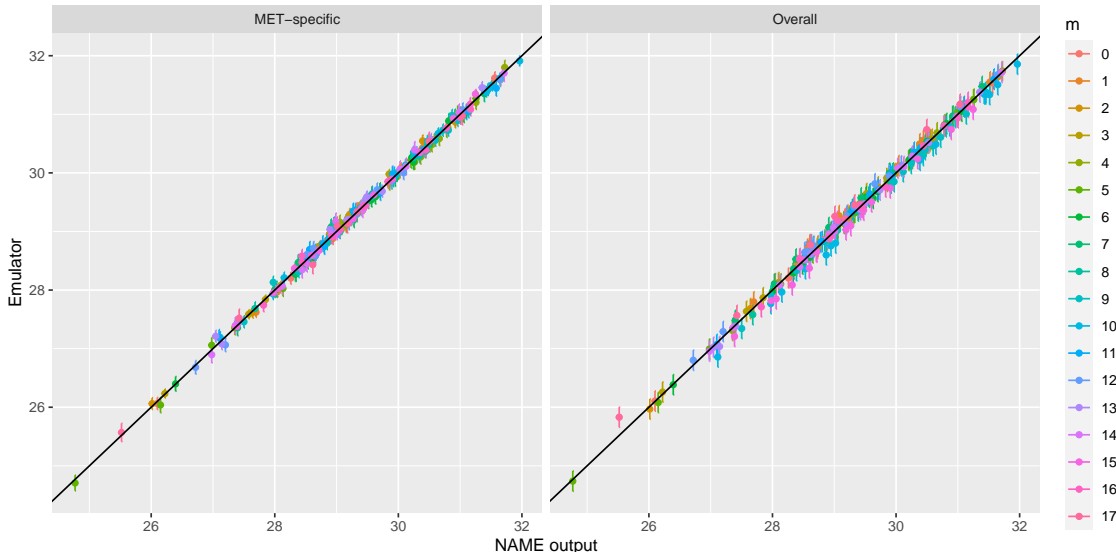

**Figure 5.** Out-of-sample emulator predictions (expectation and 95% interval) plotted against NAME output for (log) total ash column load at T1, for the MET-specific (left) and overall (right) emulators, coloured by meteorological ensemble member.

The results are similar when emulating the different regional splits at T1 (see Appendix): the MET-specific emulators perform accurately for the North/South and West/East splits, and although they become less accurate at capturing the mean for the split into 4 regions, there is still high predictive ability. The overall emulators validate similarly for the West and East regions (agreeing with the earlier assessment that this split did not introduce much meteorological dependence), however there is evidence of greater effect of $m$ with the North/South split, and even more so when splitting into 4 regions instead.

Overall, the MET-specific emulators show that when we account for the meteorological dependence explicitly, then we can accurately predict the total ash across observed locations, despite only training these emulators on a small number of inputs **x**. At longer lead times, or considering smaller geographical regions at T1, the overall emulator suggests that the meteorological variability is increasing, however according to the MET-specific emulators, the joint dependence between **x** and $m$ is generally retaining its predictability. This gives us a tool that can be applied to rapidly produce predictions across all meteorological ensemble members for any particular choice of **x**.

## 4.3 Predicting across meteorological scenarios

Given the emulators are demonstrating predictive ability, these can be used to explore questions that are time-consuming if using only NAME. Here, we evaluate the emulators at particular choices of **x** across all 18 ensemble members from MOGREPS-G, explicitly giving the uncertainty in the simulator output due to $m$. We evaluate both the overall emulator and the 18 MET-specific emulators, and visualise the effect of changing the meteorological input.





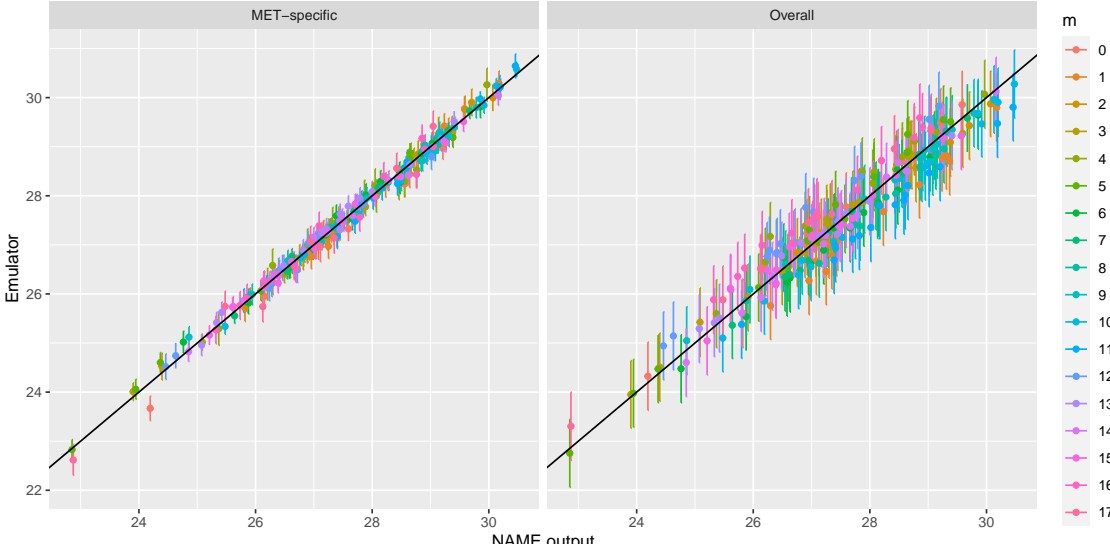

**Figure 6.** Out-of-sample emulator predictions (expectation and 95% interval) plotted against NAME output for (log) total ash column load at T5, for the MET-specific (left) and overall (right) emulators, coloured by meteorological ensemble member.

Figure 7 plots the emulator predictive distributions for the (log) total ash at T1, where $\mathbf{x}$ has been set as the first 9 input vectors used to generate the 1000-member NAME ensemble, i.e. these were simulated with NAME using a single $m$, sampled at random, and the outcome of this simulation is indicated by the vertical black line in each panel. For the total at T1, the

emulator predictions are relatively consistent across the 18 choices of meteorological scenario. However, at T5 (Figure 8), there is much greater divergence in the 18 MET-specific emulator predictions at a given input $\mathbf{x}$, again illustrating there is higher variability at T5 due to $m$.

More specifically, these plots highlight potential issues with calibration of the input parameters $\mathbf{x}$. Considering panel 7 in Figure 8, the log(total ash) value given from simulating NAME at this design point and a sampled $m$, is $\approx 27.6$ (vertical

black line), which is away from the observed value (dashed vertical line, $\approx 26.3$). Based on this comparison alone, we'd likely conclude that this choice of $\mathbf{x}$ does not lead to NAME output that is consistent with reality. However, we see that there are 2 or 3 predictions for different choices of meteorological input that are shifted more towards the observation, and that substantially overlap with this - indicating that if we had, by chance, sampled a different $m$ at which to run this particular $\mathbf{x}$, then the NAME output may have been consistent with the observation, and we would retain this $\mathbf{x}$.

This example is not unique to this particular panel and this output, and it is clear that the particular $m$ at which NAME was simulated for a given $\mathbf{x}$ may give a misleading representation or conclusion about that input. Particularly for T5, we see that the variability due to $m$ is large relative to the emulator uncertainty at a particular $\mathbf{x}$, and often only a small number of the distributions might intersect with the observations. The particular $m$ we simulated NAME at may be a biased version of what we would find if we were able to run NAME at $\mathbf{x}$ for all 18 choices, but we will not know this by only conducting NAME runs





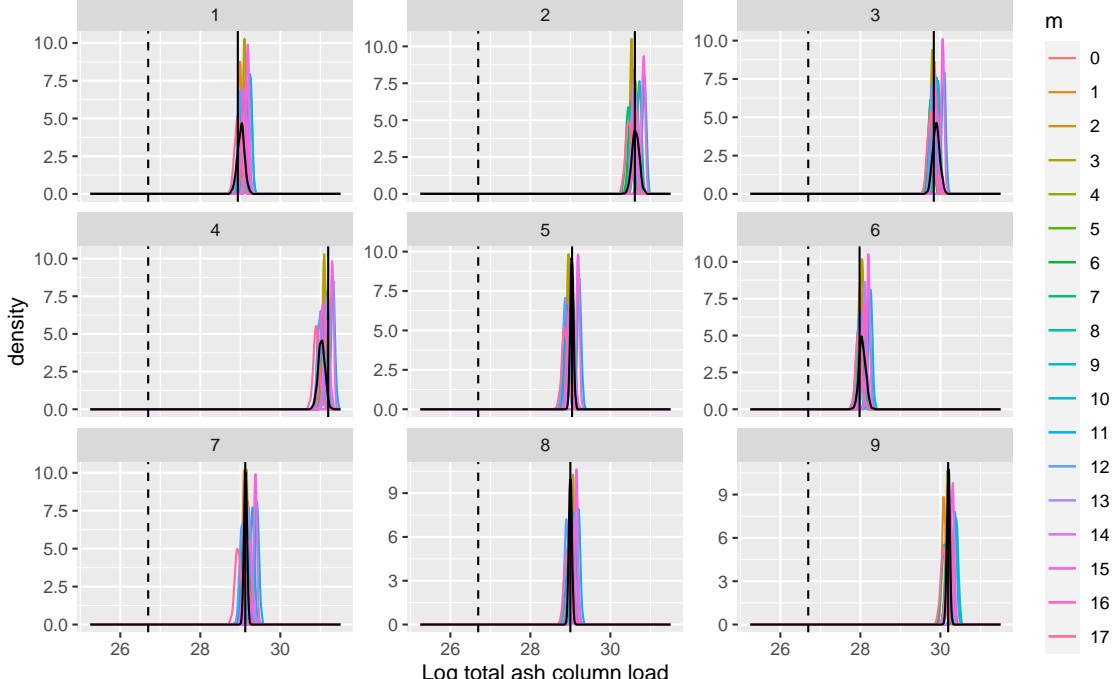

**Figure 7.** Emulator posteriors for (log) total ash column load at T1 evaluated across all 18 meteorological ensemble members (different colours) and for the overall emulator (density given by black curve), for the 1st 9 design points $\mathbf{x}_1, \ldots, \mathbf{x}_9$ in the 1000-member ensemble. The vertical solid black line indicates the value given when NAME was run at each of these 9 parameter settings (for a randomly-sampled meteorology). The vertical dotted black line indicates the observed (log) total ash at T1.

at $\mathbf{x}$ for a single $m$. However, using the emulators, instead of a single estimate given by running NAME at a particular $m$, we can now predict for each $m$, and produce a distribution for the output at input $\mathbf{x}$, across all $m$.

### 4.4   History matching to pseudo observations

In this section, we use each of the 250 NAME simulations from the validation set as 'pseudo observations', i.e. we set $z = f(\mathbf{x}^*)$ for a known $\mathbf{x}^*$. We then assess whether the different emulators for NAME, and different definitions of NROY space (Section

3.3) are able to accurately identify this true input $\mathbf{x}^*$. As these simulations are taken from the validation set, in each case we are considering inputs different from what the emulators have been trained on. We consider the different emulated summaries from earlier, and vary the observation error variance $\mathrm{Var}[e]$ (more a 'tolerance to error' here). Using the emulator expectation and variance at $\mathbf{x}^*$ for each choice of pseudo observations $z$, we can calculate $\mathcal{I}(\mathbf{x}^*)$ and $\mathcal{I}_m(\mathbf{x}^*)$, and consider the different definitions of NROY space.

We would always want the true $\mathbf{x}^*$ to be within NROY space - occasionally this will not be true (some predictions always lie outside of 95%, likewise 99%), and it may fail if emulator prediction is biased. A large NROY space is not necessarily





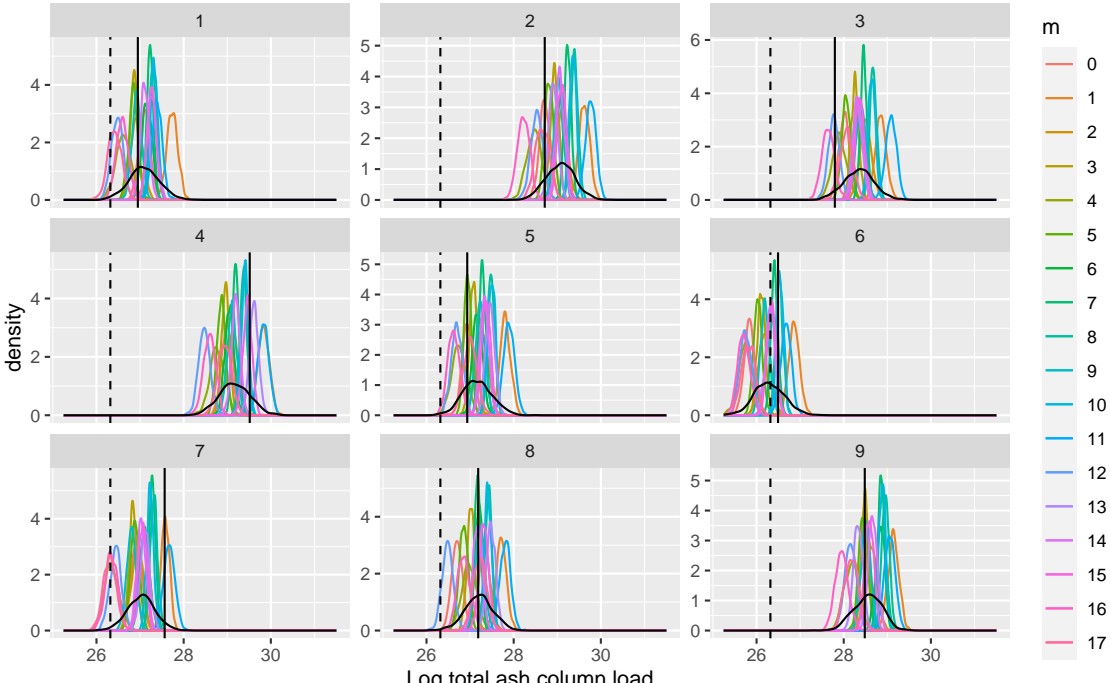

**Figure 8.** As Figure 7, comparing emulator predictions with the NAME output and observed value, but for T5.

bad - it can signify either that our tolerance to error is large, that our emulator is relatively uncertain, or that large parts of parameter space do in fact lead to simulator output 'close' to the observations - and may in fact be a realistic representation of the uncertainties in the inputs and observations.

For each choice of pseudo observations, as a simple metric, we check whether the assumed truth $\mathbf{x}^*$ is retained, or do we incorrectly rule it out? This doesn't give a complete picture, as perfect accuracy would be achieved by never ruling out any inputs. If emulator variance and/or observation error are large relative to the range of NAME output, 100% of possible inputs being considered 'close enough' to the truth could be correct, but neither is true here. To provide additional information for each choice of $z$, we also calculate the percentage of the remaining 249 validation points that are not ruled out in each case,

and report the median size of this space across the 250 different choices of $z$ (to demonstrate that any perceived accuracy is not being driven by falsely retaining all choices of $\mathbf{x}^*$).

    Even whilst knowing the input that generated the pseudo observations, we cannot perfectly assess the accuracy of the different methods: we don't know the 'true' size of each space, i.e. what percentage of input vectors would lead to output consistent with each choice of pseudo observations, up to observation error. To do so would require evaluating an $\mathbf{x}$ at all

$m \in \mathcal{M}$ to know whether this $\mathbf{x}$ is ruled out, across large samples of $\mathbf{x}$, and this is not practically feasible with NAME.

    We briefly outline the expected results for each definition of NROY space:





**Table 2.** For a given NAME output, a choice of observation error, and definition of NROY space, how often the 'true' **x** is incorrectly ruled out when the 250 validation points in turn are taken to be the pseudo observation. In each box, the entries are i) the raw number of misclassifications out of 250 and ii) median percentage of space that is not ruled out in each experiment.

| NAME output | Obs error | Pseudo $\mathcal{X}_P$ | Overall $\mathcal{X}_O$ | Cons. $\mathcal{X}_C$ |
|---|---|---|---|---|
| T1 | 1 | 0, 47% | 0, 48% | 0, 54% |
|  | 0.1 | 0, 18% | 0, 20% | 0, 26% |
|  | 0.01 | 11, 10% | 1, 14% | 0, 19% |
| T3 | 1 | 4, 29% | 0, 40% | 0, 48% |
|  | 0.1 | 42, 15% | 3, 31% | 0, 34% |
|  | 0.01 | 64, 11% | 3, 30% | 0, 33% |
| T5 | 1 | 21, 27% | 0, 48% | 0, 53% |
|  | 0.1 | 79, 16% | 0, 44% | 0, 43% |
|  | 0.01 | 92, 14% | 0, 43% | 0, 42% |
| T1 N+S | 1 | 22, 27% | 7, 54% | 0, 45% |
|  | 0.1 | 182, 1% | 10, 43% | 0, 19% |
|  | 0.01 | 250, 0% | 10, 41% | 0, 12% |
| T1 W+E | 1 | 0, 42% | 0, 44% | 0, 50% |
|  | 0.1 | 5, 13% | 2, 17% | 0, 22% |
|  | 0.01 | 38, 6% | 8, 11% | 0, 15% |
| T1 4 regions | 1 | 69, 14% | 11, 49% | 1, 37% |
|  | 0.1 | 244, 0% | 12, 37% | 1, 15% |
|  | 0.01 | 250, 0% | 13, 36% | 7, 7% |
| T1 At least 3/4 | 1 | 8, 38% | 2, 62% | 0, 55% |
|  | 0.1 | 104, 5% | 5, 52% | 0, 29% |
|  | 0.01 | 239, 0% | 8, 50% | 0, 20% |

1. We don't evaluate $\mathcal{X}_S$ here, as it uses NAME itself, no emulator prediction is required, and hence we always retain $\mathbf{x}^*$. We also can't evaluate whether we would retain this $\mathbf{x}^*$ if we'd use a different $m$ (no repeats in the NAME runs). Given





accurate emulators, $\mathcal{X}_P$ should give a reasonable approximation of what $\mathcal{X}_S$ would look like if we could run many more
simulations of NAME.

2.  For $\mathcal{X}_P$, we calculate $\mathcal{I}_m(\mathbf{x})$ using the MET-specific emulators. If variability across $m$ is low (relative to other uncertainties) this should accurately retain $\mathbf{x}^*$, however if $m$ has a larger effect, we might incorrectly rule out $\mathbf{x}^*$: we are effectively only allowed to view the emulator at a single $m$, representing running NAME once per $\mathbf{x}^*$ at a uniformly-sampled meteorological input, which may bias inference about the inputs.

3.  For $\mathcal{X}_O$, we evaluate the overall emulator and explore $\mathbf{x}$ with $m$ uncertain but accounted for, which should identify $\mathbf{x}^*$ if the meteorological uncertainty is being properly captured. The expectation of this emulator is unlikely to be consistent with the particular $m$ that we used to generate $z$, and we would expect that sometimes the truth, $z$, lies on the edge of the posterior distribution for $f(\mathbf{x})$, because we happened to view $\mathbf{x}^*$ at an 'extreme' choice of $m$, and so we may sometimes rule out this input.

4.  For $\mathcal{X}_C$, we evaluate the MET-specific emulators and jointly explore $(\mathbf{x}, m)$. This space should contain $\mathbf{x}^*$ if the emulators are accurate - it doesn't matter if $z$ was defined using a particularly extreme $m$, e.g. where 17 out of 18 meteorological scenarios would lead to $f(\mathbf{x}^*)$ being far from $z$, because we are able to explicitly consider all combinations, and only require $f_m(\mathbf{x}^*)$ to be considered 'close enough' to the pseudo observations for a single $m$.

It is important to note that $\mathcal{X}_P$ and $\mathcal{X}_C$ use the same emulators, so that any differences in accuracy between these two is not
emulator-driven, but due to the different treatment of the meteorological scenario (sampled uniformly vs fully varied).

Table 2 shows the number of times we fail to retain the true $\mathbf{x}^*$, across the 250 choices, for different NAME outputs and definition of NROY space, with varied observation error. The value in the 'Obs error' column refers to a scaling factor applied to the observation error variance $\mathrm{Var}[e]$ for the true observations. As the pseudo observations are exactly equal to NAME output and so there is no observational error here, this can be viewed as decreasing our tolerance to error - can we find simulations
closer to $z$, and do we still find the true $\mathbf{x}^*$ that generated the observations as we decrease this tolerance? N+S and W+E refer to requiring the implausibility for both emulated regions to fall below $T$ in order to consider a point not ruled out. Similarly, '4 regions' requires a point to be retained for the NW, NE, SE, and SW regions simultaneously. 'At least 3/4' relaxes this, allowing 1 region to be considered implausible (to protect against a single biased emulator causing misclassification of inputs).

In every case, $\mathcal{X}_C$ performs the best in terms of retaining the true $\mathbf{x}^*$, and in most cases never rules out $\mathbf{x}^*$, even as the error
tolerance decreases. This is unsurprising, as the emulators are reasonably accurate out-of-sample, and this definition only rules out a parameter setting if we are confident enough that it could not be consistent with the truth, for any of the 18 scenarios.

At the other end of the scale, $\mathcal{X}_P$ almost always performs the worst, and often fails to find the true $\mathbf{x}^*$. It is accurate in situations where the meteorological variability is lowest (T1 total, T1 W+E), but is poor once error tolerance reduces (e.g., even for T1, once we reduce $\mathrm{Var}[e]$ enough, it starts to fail), or variability across $m$ increases (e.g. for T5, N+S split, 4 region
split). Again, this result is expected - for $\mathcal{X}_P$ to identify $\mathbf{x}^*$, it requires consistency with $z$ across at least half of the choices of



$m$. This is a proxy for $\mathcal{X}_S$, and shows that even if the true observation was generated by some known $\mathbf{x}^*$ and $m$, we would often fail to identify this $\mathbf{x}^*$ without accounting for the meteorological variability, and we risk incorrectly constraining the inputs.

$\mathcal{X}_C$ and $\mathcal{X}_O$ generally lead to a larger percentage of the input space being retained, but it is difficult to know whether this is correct without a large number of NAME simulations. However, across different summaries and varied $\text{Var}[e]$, we see that we
rarely or never rule out the chosen $\mathbf{x}^*$, even as the tolerance to error decreases, and when $\mathcal{X}_P$ performs very poorly (suggesting larger meteorological variability relative to observation error and emulator variance). Even if bad parameter settings are not being ruled out often enough in these cases, this result is still preferable to over-constraining $\mathcal{X}$ and ruling out the 'correct' $\mathbf{x}^*$.

Some of the examples however give strong evidence that the results are as intended: for the N+S split, even with the largest observation error, the true $\mathbf{x}^*$ is ruled out by $\mathcal{X}_P$ in 22 of 250 experiments, and the median size of the retained parameter space
across these 250 is 27%. Even when $\text{Var}[e]$ is 100 times smaller, $\mathcal{X}_C$ never rules out the true $\mathbf{x}^*$, with a median of 12% of all inputs retained: more space is being ruled out, to within a lower tolerance to error, whilst still more accurately identifying the correct answer, thanks to evaluating all combinations of $\mathbf{x}$ and $m$.

## 4.5 History matching to satellite observations

Having demonstrated the accuracy of emulation and performance of the history matching definitions in an idealised setting,
we now set $z$ as the satellite observations. We have the observation error variance as before, and search for $(\mathbf{x}, m)$ such that the NAME output is within observation error of reality. Table 3 gives the percentage of input space that is not ruled out for different combinations of the emulated outputs for each NROY definition. Unlike in the previous section, we can consider $\mathcal{X}_S$ here, because we do not know the true input, and can compare distributions derived from NAME to emulator-based ones.

As expected, $\mathcal{X}_S$ and $\mathcal{X}_P$ usually have a similar size: the definition of $\mathcal{X}_P$ is such that it approximates $\mathcal{X}_S$, with emulation
rather than only NAME simulations. Some of the differences between these two are driven by the relatively small number of samples (1000) used to define $\mathcal{X}_S$, but the general similarity suggests that emulator variance is not high relative to other variability (observation error, across different inputs), and that replacing NAME with an emulator is not causing substantially more space to be retained.

$\mathcal{X}_C$ always retains a higher percentage of the input space, and this uses the same set of emulators as $\mathcal{X}_P$, hence the increase
in the percentage of $\mathcal{X}$ being retained is being caused by the definition of NROY, not by emulator uncertainty. This gives some evidence that the larger percentages of space retained by $\mathcal{X}_C$ may in fact be reasonable, and is due to the fact that exploring more combinations $(\mathbf{x}, m)$ leads to the identification of substantially more possibilities for $\mathbf{x}$.

Conversely, this suggests that there's a non-zero probability that $\mathcal{X}_P$, and hence the simulator-only approach $\mathcal{X}_S$, rules out choices of $\mathbf{x}$ that would in fact be considered not implausible if we happened to have simulated NAME at that $\mathbf{x}$ with a different
sample of $m$. In some cases (e.g., the T1 total, and the 'West' region), there is not much difference in the sizes of the 4 spaces, which suggests that the meteorological variability is low (compared to other uncertainties) for these metrics. However, even in the most similar case (T1, with 13.5% for $\mathcal{X}_C$, 11.7% for $\mathcal{X}_S$), 1.8%, or 18 of the 1000 runs, may have matched reality (up to observation error) if simulated at an alternative $m$. As in the previous section, if we consider summaries with more variability across choices of $m$ (e.g., South region), the sizes diverge more, adding evidence that if we consider more local aspects of the





**Table 3.** Percentage of parameter space $\mathcal{X}$ that is not ruled out yet for the 4 different definitions of NROY space, for different individual summaries and combinations thereof. In each row, the lowest and highest percentages are highlighted by underlining and bold text respectively.

| Summary/region | Sim. $\mathcal{X}_S$ | Pseudo $\mathcal{X}_P$ | Overall $\mathcal{X}_O$ | Cons. $\mathcal{X}_C$ |
|---|---|---|---|---|
| T1 | 11.7 | 10.9 | 11.1 | **13.5** |
| T3 | 11.4 | 11.3 | 16.4 | **26.7** |
| T5 | 16.8 | 23.0 | 40.7 | **45.5** |
| T1 + T3 | 8.3 | 8.6 | 9.6 | **12.3** |
| T1 + T3 + T5 | 0.9 | 1.3 | 5.6 | **8.6** |
| North | 17.8 | 17.4 | 31.7 | **47.0** |
| South | 3.8 | 2.5 | 7.6 | **13.6** |
| North + South | 2.4 | 1.4 | 6.7 | **12.9** |
| West | 8.0 | 8.1 | 8.1 | **10.8** |
| East | 15.8 | 16.7 | 17.2 | **22.7** |
| West + East | 7.0 | 7.1 | 7.2 | **10.1** |
| NW | 14.5 | 14.0 | 28.1 | **46.0** |
| NE | 23.1 | 26.8 | 43.0 | **64.0** |
| SE | 5.0 | 4.1 | 7.2 | **12.7** |
| SW | 3.1 | 2.0 | 9.1 | **17.4** |
| All 4 | 0.7 | 0.4 | 5.0 | **11.1** |
| At least 3 | 2.9 | 2.7 | 8.8 | **17.1** |

NAME output, the impact of the meteorological scenario increases, and jointly assessing the effect of **x** and $m$ is critical to ensure accurate understanding and calibration of the inputs.

The percentage of input space $\mathcal{X}$ that is retained varies substantially depending on the choice of NAME output, and choice of NROY definition, and observed sensitivities and relationships between different inputs are also dependent on these choices. Figures 9 (log(MER)) and 10 ($m\sigma U$) plot the prior and posterior distributions for these two inputs (averaged across the other

inputs), across different NAME outputs and definitions of NROY space.

In each case, the distribution of log(MER) has been constrained substantially compared to the prior. For the T1 total and the split into W+E regions, the 3 posteriors are relatively similar, however for the split into N+S and into 4 regions, which (from earlier) have a larger meteorological effect, the posteriors are different, with $\mathcal{X}_P$ resulting in much narrower distributions than





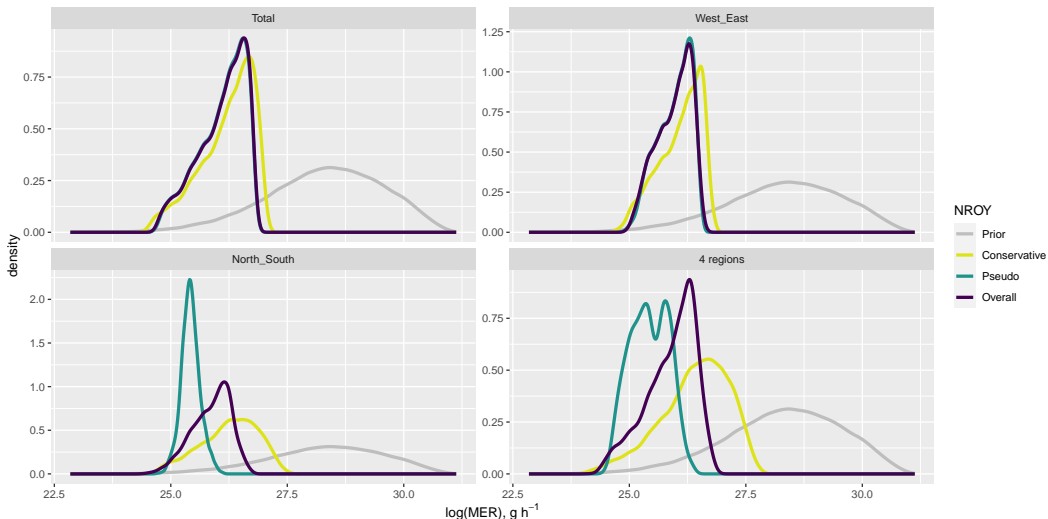

**Figure 9.** The distribution of log(MER), for different summaries and definitions of NROY space, compared to the prior.

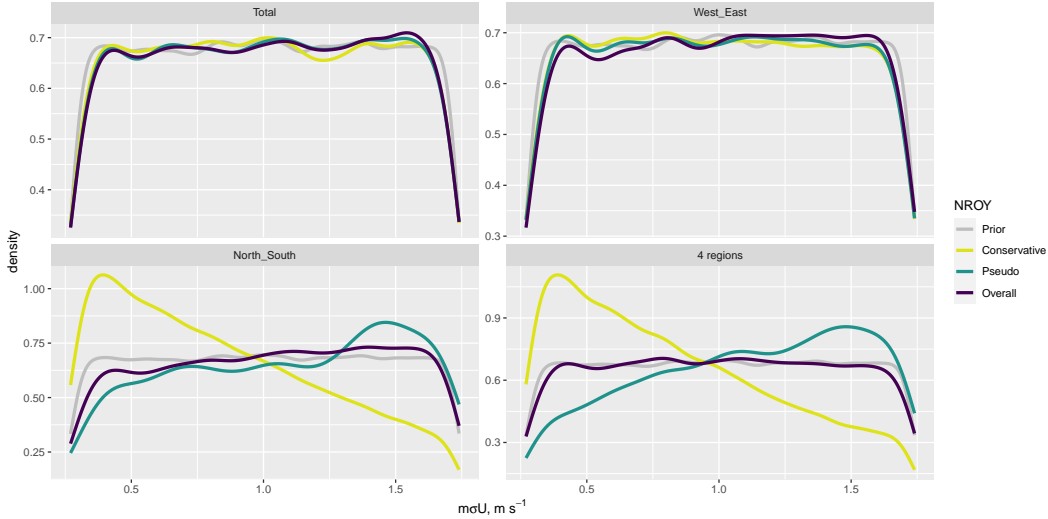

**Figure 10.** The distribution of $m\sigma U$, for different summaries and definitions of NROY space, compared to the prior.

the other definitions, whilst being contained within the other distributions. $\mathcal{X}_C$ shows that higher values of log(MER) may be

possible, given the 'correct' choice of $m$.

For $m\sigma U$, considering the T1 total or W+E split has little effect compared to the prior, regardless of NROY definition, however there are clearly different results when considering the other regional splits. For some outputs, we might conclude that this input is not affecting the output, however if we consider more localised summaries, and fully account for meteorological uncertainty, there is a strong preference for lower values of $m\sigma U$. Figure 11 shows the pairwise relationship between log(MER)




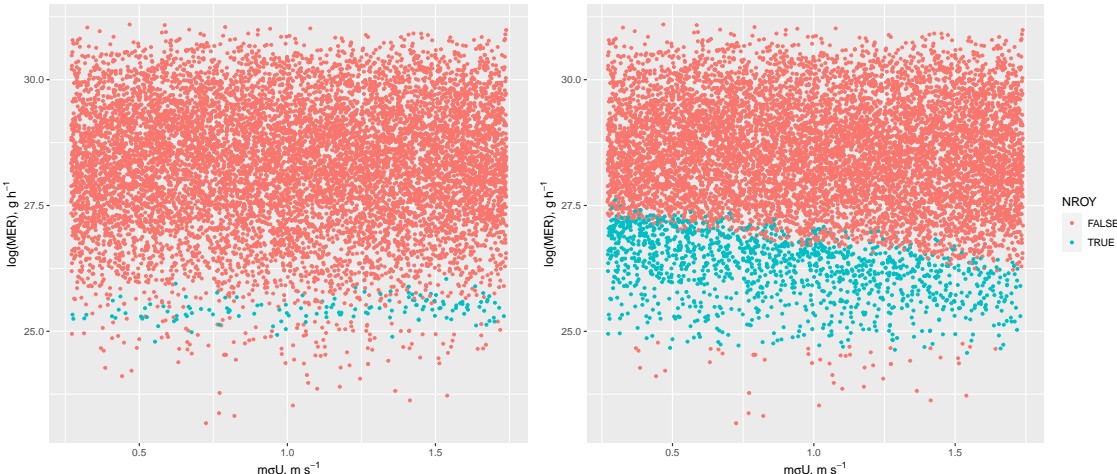

**Figure 11.** Pairwise representation of which combinations of log(MER) and $m\sigma U$ are in the $\mathcal{X}_P$ (left) and $\mathcal{X}_C$ (right) NROY spaces, when history matching for the North and South regions at T1.

and $m\sigma U$ for the North/South regional split, for $\mathcal{X}_P$ and $\mathcal{X}_C$. The opposite relationship observed for $m\sigma U$ in the 1D plot (Figure 10 is due to an interaction with possible values of log(MER): $\mathcal{X}_C$ allows higher values of log(MER) to be considered not implausible, if combined with a relatively low value of $m\sigma U$.

## 5 Discussion

In this paper, we have emulated different summaries of NAME output for the Raikoke 2019 eruption, simultaneously exploring the dependence on, and uncertainty due to, the eruption source inputs, the internal NAME parameters, and the meteorological scenario. We demonstrated the accuracy of these emulators, and showed that we can efficiently make predictions of the NAME output for any combination of $(\mathbf{x}, m)$ (with uncertainty due to this being an approximation of NAME). We used these emulators to fully explore the effects and interactions that the different inputs (source, internal, meteorological) have on the output, accounting for the uncertainties in each of these.

Given emulators, we calibrated the source and internal NAME parameters via history matching, using different outputs, emulator type, and definition of the retained (NROY) part of the input space. This demonstrated that randomly sampling the meteorological scenario (equivalently, just running NAME) leads to a much more constrained input space, compared to only ruling out combinations of the source and internal inputs if they would be inconsistent with satellite observations for all 18 possible meteorological scenarios. Due to the expense of the simulator, such a comparison is not usually feasible, however with emulators accounting for meteorological uncertainty, this can be done efficiently, with the aim of ensuring we do not incorrectly constrain the source and internal inputs due to biases from only viewing NAME at $\mathbf{x}$ for a single $m$.



For each emulated output, using a more conservative definition of the not ruled out yet space unsurprisingly resulted in a larger percentage of the original parameter space being considered not inconsistent with observations. However, the use of different choices of pseudo observations demonstrated that this appears to be more accurate, and we also showed that this larger space was not being driven by emulator uncertainty, but by the uncertainty due to the meteorological input. When the meteorological effect is smaller (e.g., as for the total at T1), the different choices of NROY space are more similar. Moving further ahead in time, or splitting the output into smaller regions, the impact of the meteorological scenario increased, resulting in a larger difference in the calibration results.

The choice of NAME summary, definition of NROY space, and how the meteorology is captured, clearly affects the conclusions we make about the importance of inputs. For example, considering only the total ash, it may appear that $m\sigma U$ is not important, however when considering ash north and south of $48.1°$N, this parameter does have an effect - and whether its distribution is skewed left or right depends on how we are accounting for meteorological uncertainty. Some differences may be driven by our choice of summary (could use other splits than at latitude $48.1°$N), however it is probable that similar differences would be evident for other outputs.

When the meteorological scenario is restricted to a small meteorological ensemble, then accurate emulators can be constructed despite a small number of training runs. However, if there are too few runs available per scenario (either if there are fewer overall NAME simulations performed, or if there is a larger available ensemble of meteorological scenarios), then treating NAME as stochastic is appropriate. Although here this emulation approach and the resulting NROY definition didn't capture the meteorological uncertainty as well as the MET-specific emulators, it resulted in greater accuracy than the simulator-only approach (and the emulator-based approximation of this). The benefit of the MET-specific emulators is that they are a more realistic representation of NAME: NAME is run for a specific $(\mathbf{x}, m)$, and this emulator predicts the output at this combination, whereas the overall one cannot. If the meteorological scenarios themselves could be parametrised with a small number of inputs, however, then a wider range of scenarios could be incorporated by an emulator, whilst still being able to predict given a specific meteorological input (see e.g. Salter et al. (2022), which demonstrates jointly calibrating simulator inputs and parameters controlling spatio-temporal temperature fields).

A benefit of history matching is in its sequential nature, and this could be exploited to simplify emulation of the full plume. For example, clearly implausible combinations of inputs and meteorological scenarios can initially be ruled out based on easy-to-emulate metrics such as the total ash, and this may remove some of the wide range of behaviours observed across the full input space. Removing implausible output makes the remaining space less variable, making the plume location more consistent, and potentially enabling the spatial aspect of the plume to be emulated more easily.

Emulation may be difficult to apply in a real-time setting, as this would require unsupervised fitting of multiple statistical models and ideally requires proper validation. However, there is a benefit in assessing past eruptions, and using this information about the model inputs to enhance simulation of future eruptions, e.g., to obtain better priors on uncertain inputs for NAME simulations, and better understanding of which inputs are most important/sensitive in different situations. It may also provide insights into probabilistic forecasting, as the emulators can more rigorously explore tails of joint distributions across



the different sources of uncertainty. Rather than considering summaries of the output, instead other quantities, such as ash at a particular location over time, could be emulated.



## Appendix A: Additional plots

### A1 Validation and prediction at T3

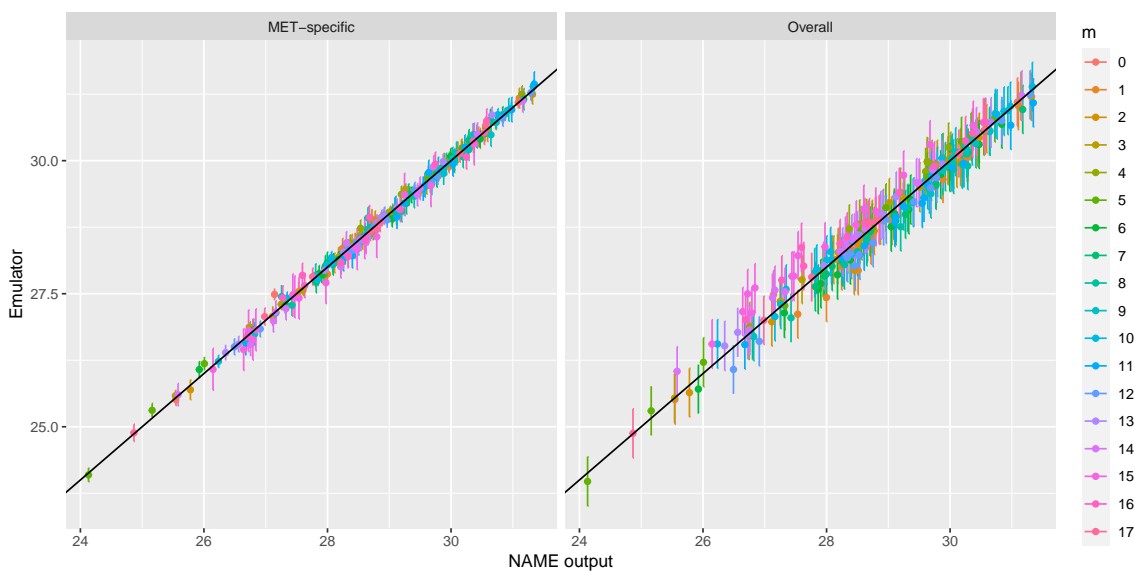

**Figure A1.** Out-of-sample emulator predictions for (log) total ash column load at T3.

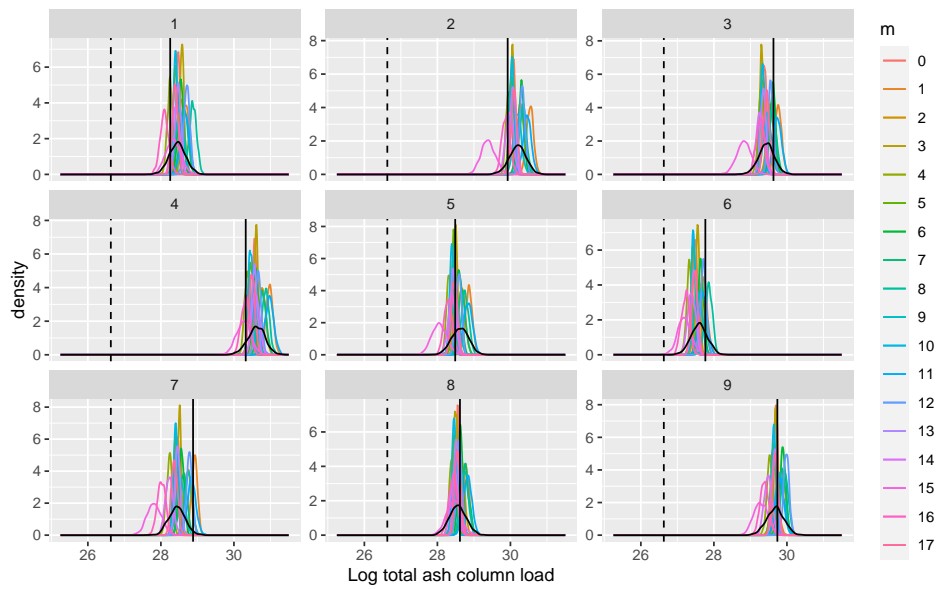

**Figure A2.** As Figure 7, comparing emulator predictions with NAME output and the observed value, for (log) total ash column load at T3.





## A2    Validation for north vs south split

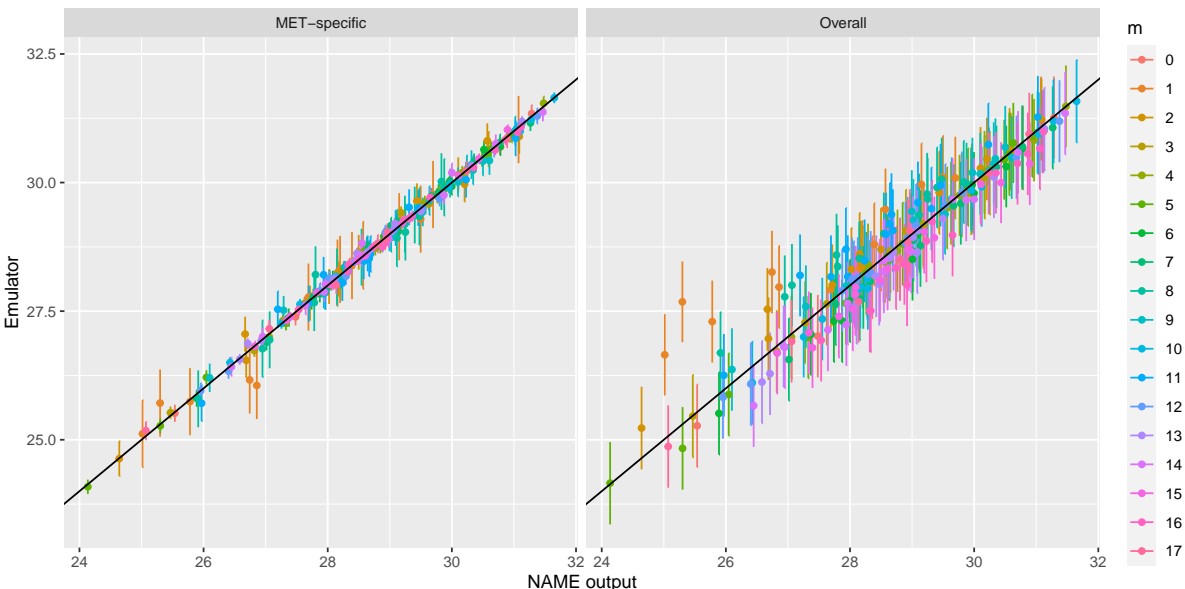

**Figure A3.** Out-of-sample emulator predictions for the 'North' region at T1

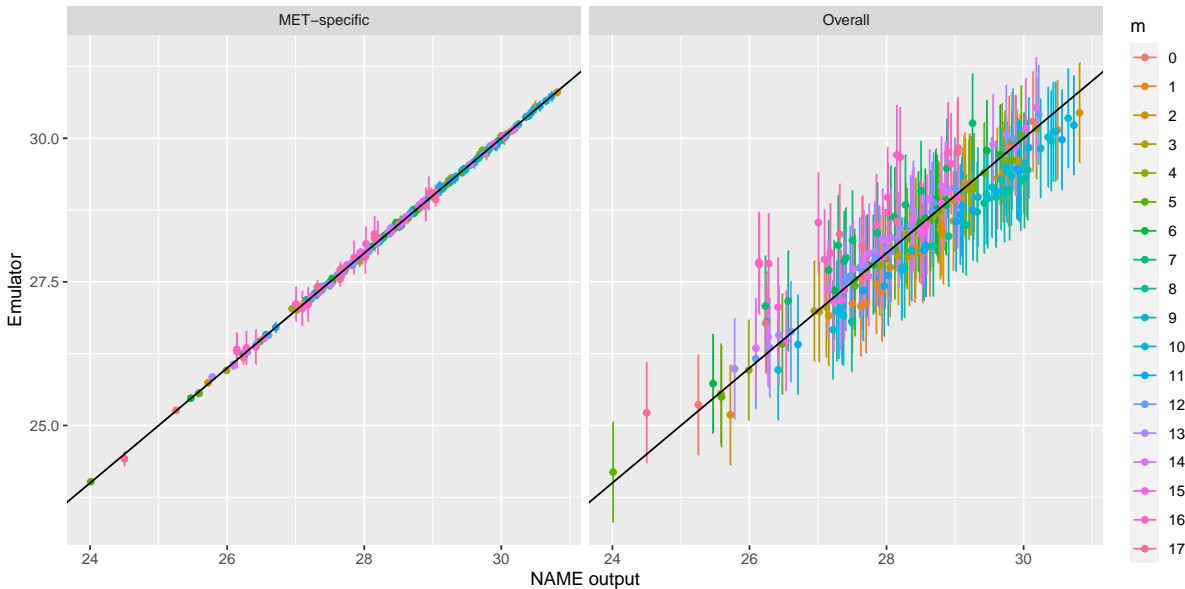

**Figure A4.** Out-of-sample emulator predictions for the 'South' region at T1




## A3 Validation for west vs east split

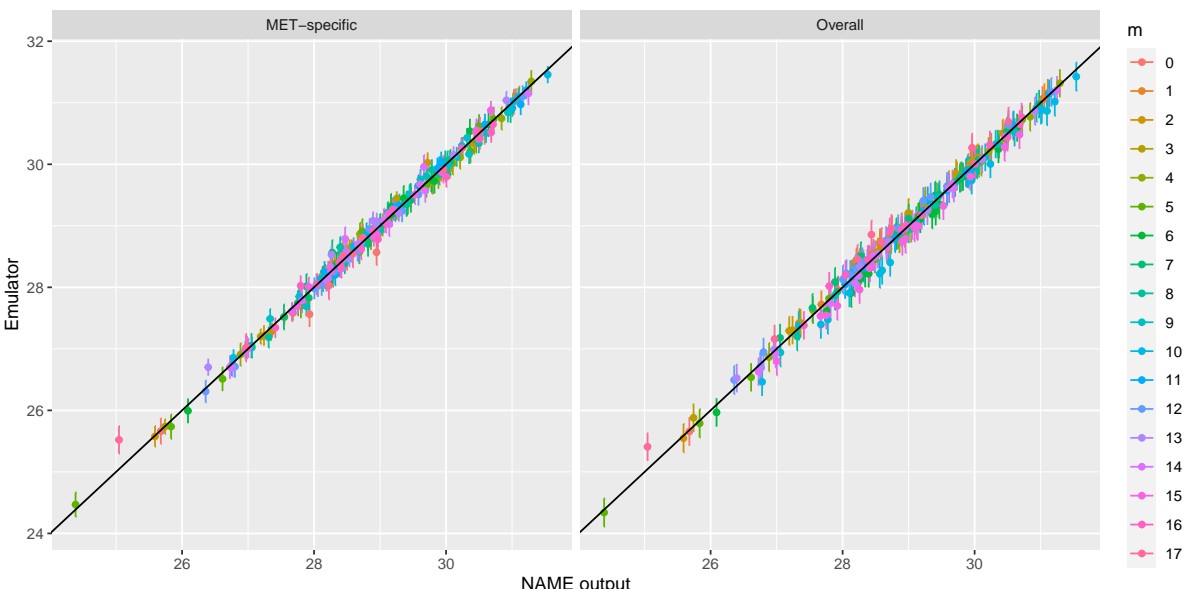

**Figure A5.** Out-of-sample emulator predictions for the 'West' region at T1

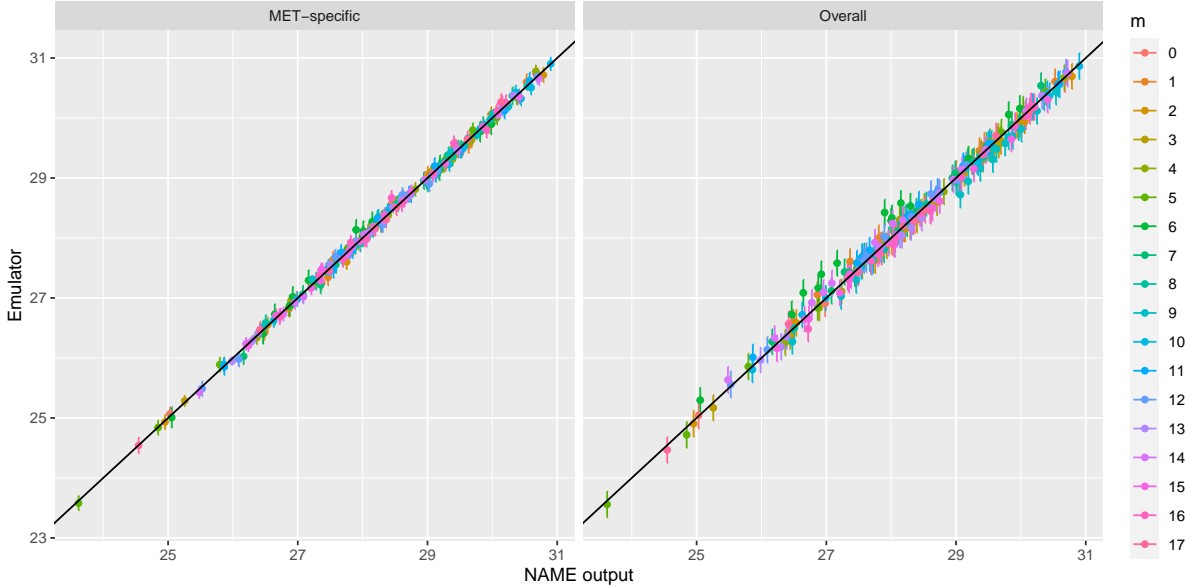

**Figure A6.** Out-of-sample emulator predictions for the 'East' region at T1





## A4 Validation for 4 region split

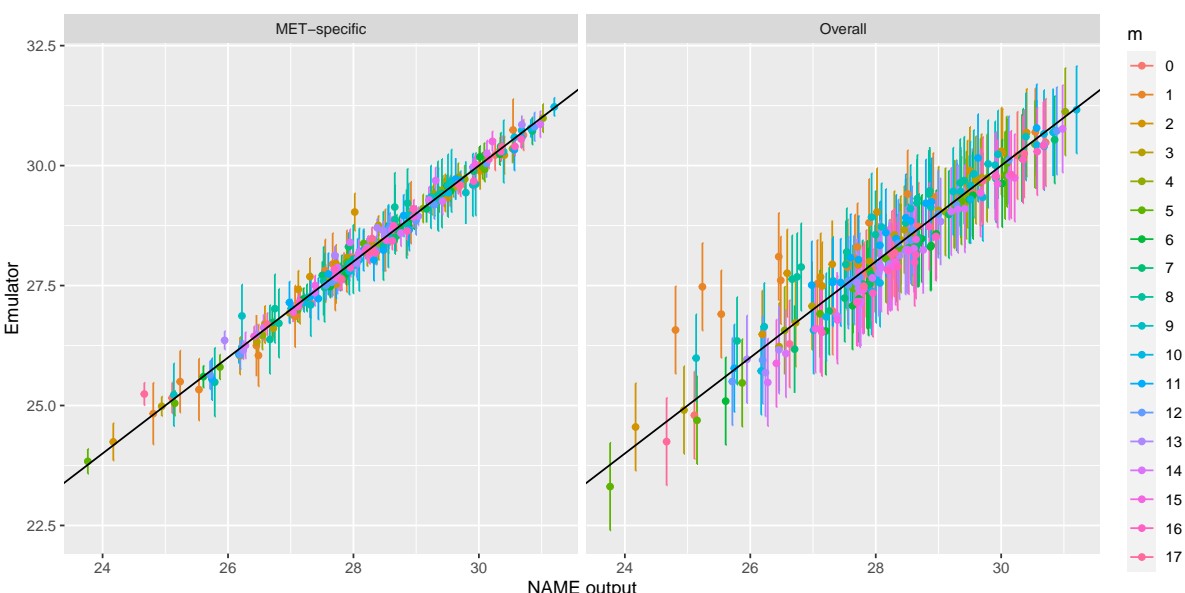

**Figure A7.** Out-of-sample emulator predictions for the 'NW' region at T1

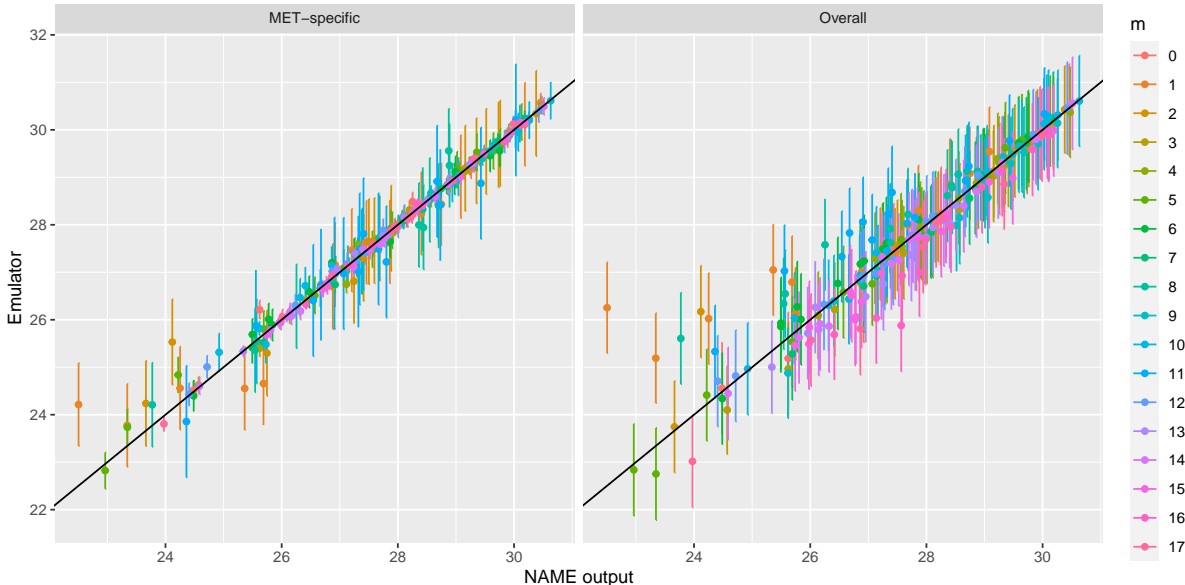

**Figure A8.** Out-of-sample emulator predictions for the 'NE' region at T1




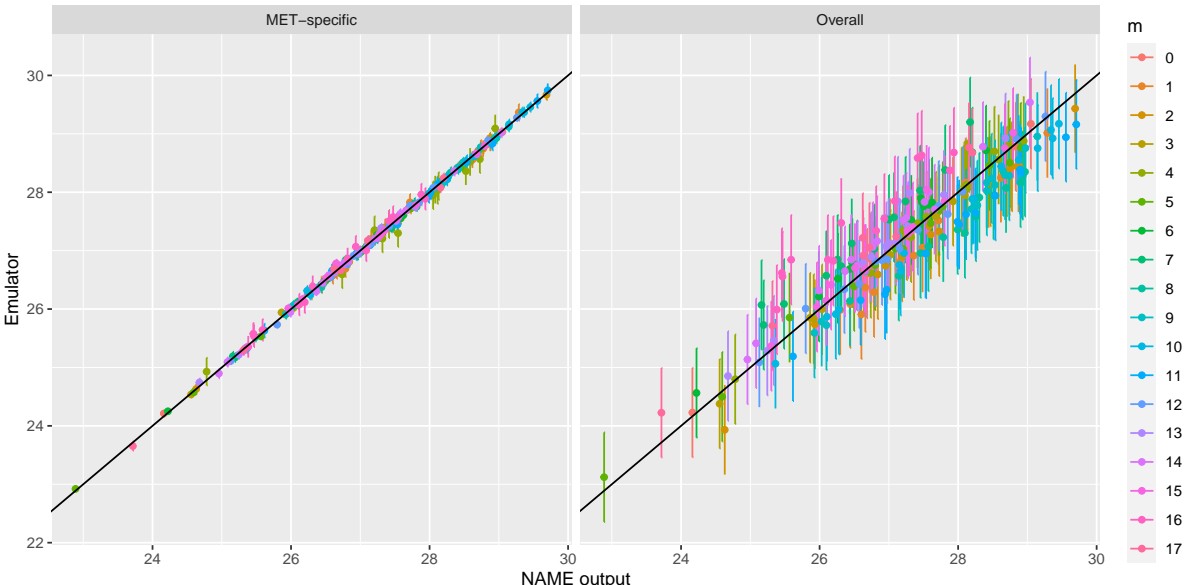

**Figure A9.** Out-of-sample emulator predictions for the 'SE' region at T1

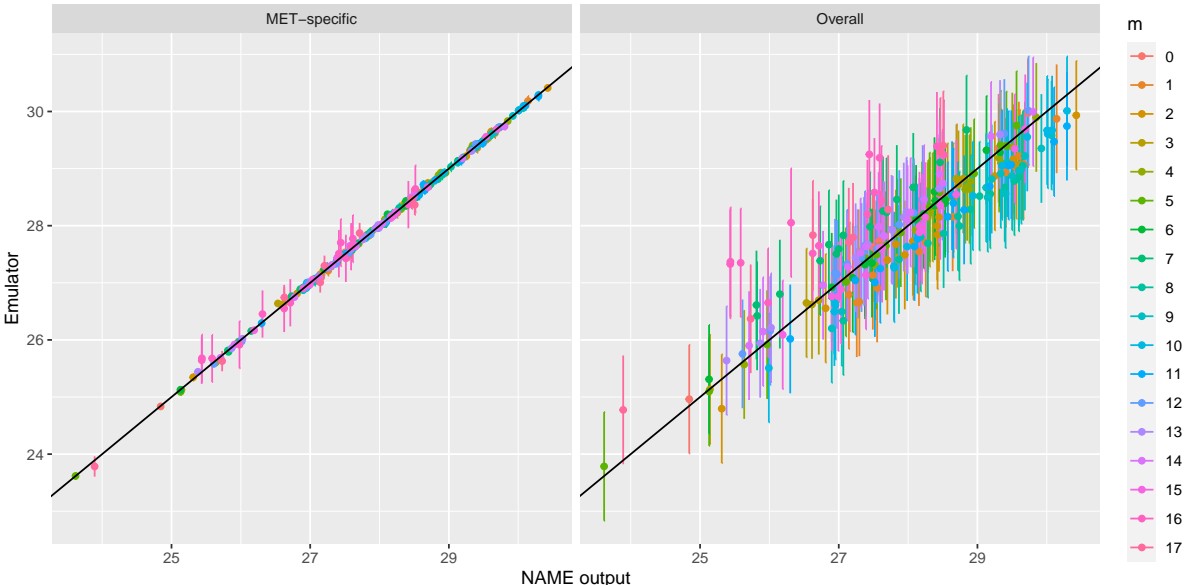

**Figure A10.** Out-of-sample emulator predictions for the 'SW' region at T1

*Code and data availability.* NAME simulation output is available at https://doi.org/10.17635/lancaster/researchdata/491 .





*Author contributions.* JS developed the statistical methodology, fitted emulators and performed the calibration experiments; HW provided
atmospheric dispersion modelling expertise to aid design and interpretation of the statistical experiments; CS provided satellite observations.
JS prepared the manuscript with contributions from all co-authors.

*Competing interests.* The authors declare that they have no conflict of interest.

*Acknowledgements.* We thank Antonio Capponi (Lancaster University) for providing the NAME simulations used in this paper.



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
