# Peer review of "Technical note: Exploring parameter and meteorological uncertainty via emulation in volcanic ash atmospheric dispersion modelling"

_EGUsphere, 2023_

## Author Response (AR1)

We thank the 2 reviewers for their helpful comments, which are addressed in turn below. The main changes have been refining the plots to make them more readable and informative, and we believe that the article is now stronger because of this and the other comments below -

**To RC1:**

**L23: I thought that probabilistic forecasts were becoming mandatory from 2025? So not just an aim but a reality?**
We've edited to make this clear:

*"There is a requirement for the VAACs to be able to provide probabilistic forecasts by late 2025"*

**L25: What is considered an expensive model in this context? Some people would think that NAME Is a cheap model!**
The main issue here was the use of 'expensive', so we've deleted this, and the rest of this section explains why to use an emulator (not possible to fully sample parameter uncertainties) without needing a judgement on what expensive means – it's context dependent, depending on what your aim/problem is.

**L25/27: In Harvey et al. they use the word simulator right from the beginning of their paper. Would that be valuable here too? (I see you do this on L113)**
Changed to:

*"When a computer model (or `simulator') cannot be"*

And also changed other instances of this in the introduction.

**L29: Can all sources of uncertainty be accounted for in the emulation?**
This was too general – rephrased as below so that this 'uncertainty' statement only relates to the emulator (and not to calibration etc. mentioned next, which accounts for other uncertainties)

*"and accounting for the uncertainty in the inputs"*

**L60: You could expand on why the meteorological scenario has a larger impact at longer leadtimes.**
Added a comment:

*"(either through the ensemble spread increasing through time, or through the meteorology affecting the spatial distribution of ash)"*

**L107: Here mention the influence of meteorological cloud and the fact that the satellite retrieval also has a detection limit (so wouldn't see the light-yellow areas on in the simulated panels in Fig 1.)**

We've edited this paragraph to clarify about the detection limit, and how this is not likely an issue in this case:

*"The observed ash cloud may be incomplete, due to failures of the detection or retrieval algorithm to either classify pixels as ash or reach an acceptable solution for the ash cloud properties. Although in some cases obscuring meteorological cloud can be an issue for the detection, most of the failures here are caused by either high ash column loadings or ash particle sizes beyond the detection algorithm's sensitivity range. Column loadings lower than approximately 0.2$g m^{-2}$ are not usually detected by this algorithm, but qualitative assessment of the satellite imagery suggests that much of the ash in the yellow regions visible from the NAME runs in Figure 1 was not present."*

**L109: How did you choose these 3 NAME simulations? Are they extremes or randomly chosen?**
Neither extremes nor random, so we've updated it to have the max/min/closest to ensemble mean as this makes more sense.

**L211: Not sure which metrics are being referred to here. Include a bit more text to set the scene.**
Rephrased so don't use metric, and been more specific what referring to:

*"Below, we outline the rationale behind each definition of NROY space we consider."*

**L263: Why were these times chosen? VAAC forecasts typically go out to 18 hours.**
This links into the comment below (L272), so both are addressed here: we've added a comment to explicitly mention VAAC forecasts going to 18 hours and justified the use of longer lead times as an attempt to see whether the emulators are still accurate as the NWP ensemble diverges more over time. Whilst this isn't a perfect proxy for studying other events where the meteorological situation is more complex in the 1st 6/12 hours, it at least shows that everything wasn't only working well because of the meteorology being relatively consistent early on, and that there is still predictability/accuracy/value from emulation as meteorological variability increases.
We've added:

*"Although VAAC forecasts only go out to 18 hours, we have used longer lead times as well to demonstrate whether the emulators retain accuracy as meteorological variability increases (other eruptions may have a wider range of meteorological forecasts in the first 18 hours than here, so this partly attempts to demonstrate whether the method can extend to more variable events)."*

**L265: Why was total airborne ash chosen as a metric? I am not sure that it is particularly relevant for decision making.**
Ash column loads were used rather than e.g., ash concentrations for comparison to observations. Totals were used a) as part of an exploration of what was predictable and b) because there is such a wide range of behaviours in the initial ensemble, simply considering totals allows a large amount of parameter space to be ruled out anyway (and removing such behaviour initially can then be beneficial for emulating the parts of the output that we do

care about). We agree that any extensions should focus on emulating metrics that are more relevant for decision making (e.g., at a location over time, as referred to in the discussion).

**L272: Here you mention assessing meteorological dependence at short leadtimes. This is relevant for the timescale used in VAAC forecasts. I also think that this can't be assessed fully using a single case study and forecast. The meteorological situation during the Raikoke eruption was reasonably straightforward – this isn't always the case!**
Added a qualifier "*for this eruption*" here as this was only supposed to refer to this case, and as you say, the NWP ensemble spread could be much higher for other events.

**L320: Presumably this is related to errors in ash location accumulating over time due to errors in the meteorology.**
There may be some effect of this, but it is not the main driver of the larger error bars. The MET-specific emulators have much smaller error bars (i.e., when conditioning on a particular m, there is still a strong, predictable relationship between the inputs and outputs). The overall emulator is capturing variability across all m for a particular input, and whilst some of this uncertainty can be attributed to predicting out-of-sample at a different x, the main driver of the difference between the left and right plots is that it is also capturing the variability across m, and if want to predict for a particular x across all m, the output is now much more variable because the NWP ensemble spread is now higher.

This is perhaps clearer in Figure 8: the dark density is the overall posterior, and this has larger uncertainty than any of the 18 MET emulators considered individually (as shown in Figure 6), but the spread across the 18 MET densities is comparable to the overall density. We've added a comment to emphasise this further:

*"The single overall emulator curve in each panel (dark line) has larger variance than any of the individual densities, as it is capturing the uncertainty across all $m$, and in each panel we see that the overall prediction is broadly capturing the spread of the 18 individual predictions."*

**L323: Is this because the areas covered by the ash plume is relatively small?**
Potentially – unclear without performing further experiments with more eruptions, but we would expect smaller clouds at the shorter lead times, and a stronger signal due to less transport errors/less NWP spread. You'd also hope that there's something predictive between the inputs/outputs regardless of the size of the ash plume / lead time. We've deleted the word "*high*" as this is subjective, whereas 'predictive ability' has been shown.

**L330: This seems very powerful but presumably case specific. How quickly could it be redone for another eruption?**
Correct, case specific as trained on the output for a particular eruption / particular met-conditions. To do so in the same way for a different eruption could be feasible (in a way that's useful for decision-making, e.g. considering flight levels, specific locations), but how quick this is, and how accurate, would depend on the particular eruption (how many simulations can do, how variable meteorology is, how wide priors are). So hard to generalise from a single event like this, and probably need to consider case studies of different eruptions with different behaviour in order to answer this properly.

We've added a clarifier to emphasise the case-specificity in this statement:

*"This gives us a tool that can be applied to rapidly produce predictions for the Raikoke eruption across all meteorological ensemble members for any particular choice of x."*

**L370/371 The use of question here makes it difficult to read. Please rephrase.**
Removed this – it now reads:

*"For each choice of pseudo observations, as a simple metric, we check whether the assumed truth x\* is retained, or incorrectly ruled out."*

**L469: Here it would be nice to give an indication of how long it takes to build/validate an emulator.**
Added a comment to this effect at the end of Section 4.1:

*"Given we already have simulations of NAME, fitting an individual emulator may take seconds (MET-specific, few training points) to several minutes (overall, 750 training points). Depending on whether the initial versions validate well, we may require repeated fittings (for example, changing the mean/covariance functions) to find a suitable emulator."*

**L490: Better use the parameter description in words rather than the symbol.**
Added.

**L502: Is it possible to parameterise the met scenario? How quickly might you be able to do that? Is it something that would just be a research tool?**
Probably challenging! Salter et al 2022 did so for 15,000 year long boundary conditions of temperature and precipitation derived from a small number of GCM/FAMOUS simulations by extracting key patterns and varying these. There should be informative patterns within the NWP ensemble that are driving where the ash ends up. Whether a general methodology can be devised so this can be done automatically is a big question, and it's not clear whether it could be done in real time or just to give a more accurate emulator for post-hoc exploration.

A clear case where there's value could be: we have fewer NAME simulations than the 1000 here, perhaps 10 simulations per NWP ensemble member. Fitting the met-specific emulators will not work as there's too few training points for each m, however using only the overall loses the ability to predict at a particular m. If the high-dimensional NWP members could be summarised by a few important features via dimension reduction techniques (to find key patterns driving variability in the output), coefficients representing these features can be appended to x, and a single emulator can now potentially predict at combinations of x and m (indirectly through these coefficients), without requiring as many simulations per NWP member.

We've rephrased this point slightly:

*"If the meteorological scenarios themselves, or key patterns from them, could be represented via dimension reduction and hence represented by a small number of coefficients, then a wider range of scenarios could be incorporated by a single emulator, whilst still being able to predict for a specific meteorological scenario"*

**L517: Ash at a particular location is much more useful for decision making. Could you speculate how emulation approaches might be able to be used in an emergency response situation here?**
We've added an extra comment here:

*"This could be approached in a similar way as in this study, with emulators trained on NAME simulations and predictions made for a more complete combination of inputs and meteorological scenarios, allowing the production and assessment of more detailed information about possible impacts at important locations."*

**Table 1: Is this taken from Capponi et al.? If it is, please include this in the caption.**
It's very similar, we've added this to the caption.

**Table 2: I initially found this a bit confusing to look at with using each cell for two metrics. Consider splitting out into two tables.**
We've split these into separate columns in the same table, so this should be much more readable now.

**Figure 1: Would this benefit from a log scale? Also, it is difficult to see the red triangle marking the location of Raikoke. Maybe this would be better as blue or green?**
Changed to log scale, and a green triangle (looked clearest in colour blindness checks).

**Figure 2: Is the colourmap used to indicate the different ensemble members colourblind friendly?**
All figures have been adapted to ensure this is the case.

**Not many members lie within the uncertainty of the satellite uncertainty. Could this be related to an upper detection limit of the satellite?**
There could be some effect of this, particularly earlier on in the eruption, although the main reason is likely that the priors on the inputs are leading to a very wide range of possible outputs. 1000 points in a 10D input space also won't necessarily sample all possible model behaviours by chance, and so in some situations we may find none of the model outputs seem to match the constraints (and this is likely true if we consider all observational constraints simultaneously – NAME is not a perfect model of reality, NWP ensembles are not a perfect representation of the weather, so no perfect (x,m) will exist, even up to observation error, and this can be handled via structural error/model discrepancy terms).

Having only a few members that do lie within the uncertainty is to some extent a function of the prior/ensemble size, and is motivation for emulation/calibration: we can sample millions of points in the input space and compare these to the observations, and figure out if we can identify the regions of the prior parameter space that lead to output closer to reality. If we were to run a new ensemble of NAME, aided by the emulation/NROY results, we should see

points that are much closer to the satellite estimates, if not within the errors, and have much less variability than the prior 1000 member ensemble.

We've added some discussion on this point:

*"Many of the members lie outside the satellite's uncertainty range. This is largely expected due to prior ranges leading to a wide range of possible eruptions, only sampling 1000 points in a 10D input space (not capturing all possible simulator behaviours), the NWP not being a perfect representation of weather conditions, and NAME not being a perfect simulator of an eruption, among others. However, it is worth noting that limitations in the satellite detection and retrieval algorithms that aren't accounted for in the provided uncertainties likely mean that the total mass present is an underestimate, perhaps by up to an order of magnitude. If this were accounted for slightly more members may be within the satellite uncertainty."*

**Figure 9: The yellow line doesn't show up very well on the grey background. Difficult to see the Pseudo line on the top two panels. This could be shown better if the Overall line was changed to dashed.**
We've made the suggested changes.

**To RC2:**

1. **Line 183: you define the variance of the observation error and model discrepancy, but don't state what values you assign these.**

These are derived from the satellite retrievals, where uncertainty is given for each pixel (described around line 280 in the updated version). To get to uncertainty for the total, it's not clear what correlation structure we should assume – likely not uncorrelated, likely not perfectly correlated. Because model discrepancy is also challenging to elicit, and because the 1000 simulations tend to lie slightly away from the observations, we assumed the errors across grid boxes are perfectly correlated, resulting in much larger 'observation error' than there should be (errors will be less than perfectly correlated), but with this also resulting in some simulations lying within these inflated error bars for each metric (at least if we consider 99% intervals). We haven't reported these variances explicitly as they're not 'real' observation errors in a way (and the value of history matching is that if we had ruled out everything for a particular metric, this tells us that we probably haven't specified our discrepancy variance properly). This didn't happen here, so this inflated error is accounting for it to some extent.

We've added:

*"As model discrepancy is unknown, we assume that this is accounted for within this inflated error variance. In history matching, if we find we rule out all of input space, this suggests we misspecified/underestimated these errors, and we should increase the discrepancy variance."*

The linked, public code/data provides the estimated observation error variances (on the log scale).

2. **Line 234: Should this be 'rule out' not 'rule'?**

Correct – changed to 'rule out'

3. **Line 291: How are the training and validation sets created? Randomly, or a stratified subset which covers the full range of input values for the most active input? (Which is the better way to subset for validation sets!)**

The only stratification was done by meteorology m, with a random set taken given m. Here, because validation didn't suggest any systematic biases in parts of the input space, doing this randomly didn't seem to matter. If validation highlighted e.g., poor extrapolation to high/low values of certain parameters, then would have reconsidered, but the random training sets seem to have had strong enough signal regardless of their design.

4. **Line 461: I found this sentence a little hard to follow.**

We've rephrased this (and the following sentence) to hopefully make this clearer:

*"For $m\sigma U$, regardless of the NROY definition, the posterior is the same as the prior for the T1 total and W+E split, and we might conclude that this input is not affecting the output. However, there are clearly differing results for the other regional decompositions, including the most localised summaries, and fully accounting for meteorological uncertainty leads to a strong preference for lower values of $m\sigma U$."*

5. **Figure 6 and corresponding appendix figures: could a percentage of coverage be added to the top left corner of the figures? It would be good to know what percentage of predictions are touching the x=y line. As errors are so small and points overlapping, it is hard to judge how many are touching the line visually.**

We've edited these plots so that it should be easier to judge this visually (colouring points green/red instead of 18 different colours overlapping). We've also edited a qualifier that

*"Across the fitted emulators, 93-97% of the true values lie within 95% prediction intervals"*.

6. **Table 2: This was a little hard to understand, perhaps could benefit from turning into two tables.**

We've split these into separate columns in the same table, so this should be much more readable now.

7. **Check if the colour maps used in the figures are colour-blind friendly.**

Some of them were not – we've refined all these as appropriate and edited the text to reflect this. In the plots where there were previously 18 different colours, they were not all individually identifiable before, so instead we've split these into panels for m=1,…16 (e.g. Figure 2,3,4), and these are now much more interpretable and informative than before.

8. **Is the code available anywhere e.g., GitHub?**

Yes, and the 'Code and data availability' statement has been edited to reflect this: https://doi.org/10.5281/zenodo.10820858

---

## Author Response (AR2)

**Both reviewers commented on the suitability of the figures for colorblind readers. Reducing the number of colors used in many of the figures certainly helps, but the choice of colors in Figures 5, 6, A3, A5-A12 appears suboptimal when I view them using a green-blind filter (e.g., https://www.color-blindness.com/coblis-color-blindness-simulator/). Specifically, the red points are difficult to distinguish from the black error bars. I request another tweak to the colors used in these plots, the colorblind-safe suggestions at https://colorbrewer2.org/#type=diverging&scheme=PiYG&n=4 may be useful.**

We've edited the colour scheme for these figures (slightly different shade of green for majority of points, dark blue instead of red for the ~5% of points, and grey error bars instead of black). These choices appear distinctive on all options on the colour blindness simulator.

We've also edited the figure captions to reflect the new colour scheme.